# SPAR: Self-supervised Placement-Aware Representation Learning for Distributed Sensing

## Abstract

We present SPAR, a framework for self-supervised placement-aware representation learning in distributed sensing. Distributed sensing spans applications where multiple spatially distributed and multimodal sensors jointly observe an environment, from vehicle monitoring to human activity recognition and earthquake localization. A central challenge shared by this wide spectrum of applications, is that observed signals are inseparably shaped by sensor placements, including their spatial locations and structural roles. However, existing pretraining methods remain largely placement-agnostic. SPAR addresses this gap through a unifying principle: the duality between signals and positions. Guided by this principle, SPAR introduces spatial and structural positional embeddings together with dual reconstruction objectives, explicitly modeling how observing positions and observed signals shape each other. Placement is thus treated not as auxiliary metadata but as intrinsic to representation learning. SPAR is theoretical supported by analyses from information theory and occlusion-invariant learning. Extensive experiments on three real-world datasets show that SPAR achieves superior robustness and generalization across various modalities, placements, and downstream tasks.

## 1 Introduction

This paper advances the state of the art in self-supervised **placement-aware representation learning**, motivated by the broad class of applications we term **distributed sensing**. By distributed sensing, we refer to systems where multiple spatially distributed sensing points—potentially spanning diverse modalities—jointly observe an environment. This definition unifies a wide spectrum of domains, including seismic and acoustic monitoring for security (Li et al., 2025; California Institute of Technology (Caltech), 1926), human activity recognition with body-worn sensors (Gu et al., 2021; Sztyler & Stuckenschmidt, 2016), vehicle monitoring in urban spaces (Bathla et al., 2022), environmental monitoring (Ullo & Sinha, 2020), and smart cities (Syed et al., 2021). These scenarios, though superficially distinct, share the common challenge of reconstructing or representing an environment from heterogeneous, distributed vantage points.

**Sensor placement** lies at the core of distributed sensing. A sensor's vantage point is determined by both its **spatial location** (e.g., GPS coordinates of a seismic station dictating which parts of the crust it samples) and its **structural role** (e.g., the body location of an IMU sensor that shapes its motion patterns). Robust representation learning in this setting requires models that not only capture signal content but also interpret how those signals are mediated by spatial and structural placement.

Despite rapid progress in sensing pretraining, current approaches—whether contrastive (Ouyang et al., 2024), generative reconstruction (Kara et al., 2024b), or language-model-based (Ouyang & Srivastava, 2024)—remain largely placement-agnostic, overlooking the fact that distributed sensing signals are inseparably shaped by sensor placement. This omission limits generalization across layouts, scales, and tasks.

To address this gap, we introduce SPAR (**S**elf-supervised **P**lacement-**A**ware **R**epresentation learning), a general-purpose pretraining framework that explicitly incorporates placement into representation learning for distributed sensing. Our design is guided by a core principle: the **duality between positions and signals**. That is, spatial and structural configurations are not auxiliary metadata to the signals, but stand in an equal and mutually-determining relationship with signals. Together, they

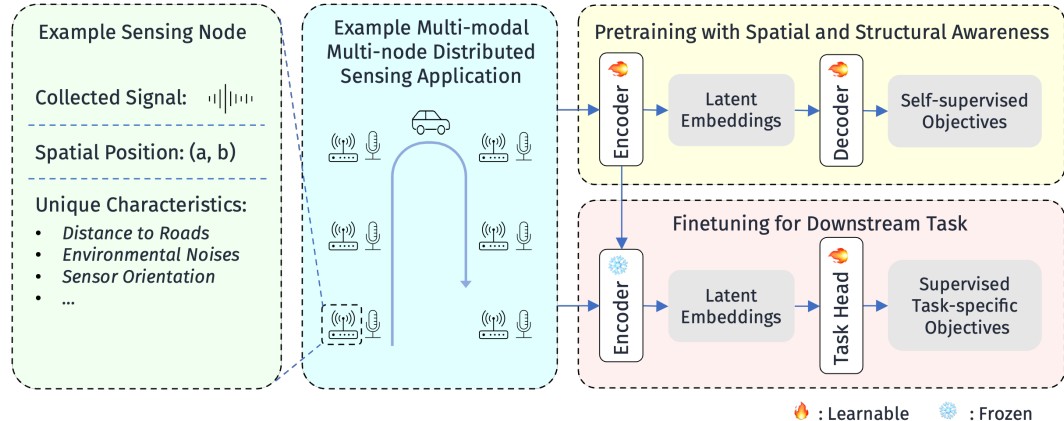

Figure 1: An overview of the SPAR workflow applied to a multi-modal multi-node distributed sensing application. Each node from each modality collects its own signal and is associated with a spatial position, as well as unique characteristics that influence its signal patterns. During pretraining, SPAR encodes information from all these aspects to generate latent embeddings, optimized via self-supervised objectives on unlabeled data. In the fine-tuning stage, the encoder is frozen and used to extract representations, which are then fed into task-specific heads trained with labeled data for downstream tasks.

define how observations are generated, propagated, and interpreted. This principle is both general and intrinsic, applying across the full spectrum of distributed sensing applications.

Building on this principle, SPAR introduces three key components: (1) **spatial positional embeddings** encoding sensor locations, (2) **structural positional embeddings** capturing node-specific characteristics, and (3) **dual reconstruction objectives** that enforce the mutual recoverability of placements and signals with contextual awareness. Together, these elements yield a cohesive, placement-aware pretraining strategy that is broadly applicable across sensing modalities and layouts. An overview of the SPAR workflow is illustrated in Figure 1. To our knowledge, this is the first work to treat placement as a universal inductive bias for distributed sensing systems as a whole, rather than as an application-specific add-on.

We further provide theoretical analyses grounded in information theory and occlusion-invariant representation learning (Kong & Zhang, 2023) to elucidate the rationale behind our design. Experiments on three real-world datasets—covering vehicle monitoring (Li et al., 2025), human activity recognition (Sztyler & Stuckenschmidt, 2016), and earthquake localization (California Institute of Technology (Caltech), 1926)—demonstrate that SPAR consistently outperforms existing approaches across diverse sensing modalities, spatial configurations, and downstream tasks.

In summary, this paper makes the following contributions: (1) We introduce SPAR, a novel, general pretraining framework for distributed sensing that explicitly models spatial layouts and node-specific characteristics, guided by the duality between positions and signals. (2) We provide theoretical analyses from information-theoretic and occlusion-invariant perspectives that explain the effectiveness of our design. (3) We validate SPAR through extensive experiments on three real-world distributed sensing datasets, demonstrating superior generalizability and robustness compared to prior methods.

## 2 RELATED WORK

**Pretraining and Foundation Models for Sensing.** Pretraining for sensing aims to learn transferable representations from unlabeled data, enabling scalable downstream learning. Existing approaches largely fall into three paradigms: contrastive learning, generative (masked reconstruction), and LLM-based frameworks. Contrastive methods align multi-modal embeddings in a shared space. Early works such as Cosmo (Ouyang et al., 2022), Cocoa (Deldari et al., 2022), and FOCAL (Liu et al., 2023) focus on intra-sample contrast via modality-specific augmentations, while more recent models like ImageBind (Girdhar et al., 2023) and MMBind (Ouyang et al., 2024) extend to loosely paired or unpaired modalities. Generative approaches rely on masked reconstruction (Woo et al., 2024; Das et al., 2024). Ti-MAE (Li et al., 2023), MOMENT (Goswami et al., 2024), and TS-MAE (Liu et al., 2025) adapt autoencoding to time series, with TS-MAE using a continuous-time formulation. Other methods, such as FreqMAE (Kara et al., 2024b) and PhyMask (Kara et al.,

2024a), introduce frequency-domain masking tailored to sensing signals. LLM-based frameworks integrate sensor data into language-centric systems (Gruver et al., 2023; Garza et al., 2023). LIMU-BERT (Xu et al., 2021) adapts masked language modeling for inertial data, while Penetrative AI (Xu et al., 2024), LLMSense (Ouyang & Srivastava, 2024), and IoT-LM (Mo et al., 2024) introduce prompting, summarization, and modality-specific adapters for cross-task transfer and zero-shot inference. While effective, these methods do not explicitly account for the spatial layout and node-specific characteristics critical to distributed sensing. In contrast, SPAR incorporates spatial and structural information directly into pretraining, enhancing contextual grounding and robustness.

**Pretraining with Different Notions of "Spatial" Context.** Several works incorporate spatial context, though definitions of "spatial" differ. In vision, it typically refers to grids of pixels or patches, as in video (Feichtenhofer et al., 2022; Wu et al., 2023a), remote sensing (Lin et al., 2023; Reed et al., 2023; Irvin et al., 2023), and 3D medical imaging (Gu et al., 2024). Beyond vision, "spatial" often denotes discrete symbolic entities, e.g., joints in SkeletonMAE (Wu et al., 2023b), EEG channels in MV-SSTMA (Li et al., 2022a) and MMM (Yi et al., 2023), or sensor identities in Gao *et al.* (Gao et al., 2023) and Miao *et al.* (Miao et al., 2024). By contrast, our method integrates continuous node coordinates into pretraining, enabling modeling of arbitrary sensor layouts that depart from token sequences or regular grids, and generalization to unseen configurations. A related but distinct line is scene reconstruction and novel view synthesis (Mildenhall et al., 2021; Kerbl et al., 2023; Wu et al., 2024), which also exploits spatial layouts but targets synthesis quality, rather than transferable representations for sensing.

**Pretraining via Positional Reconstruction Objectives.** A third line of work, often without using the term "spatial," incorporates positional reconstruction objectives. In vision, Doersch *et al.* (Doersch et al., 2015) proposed predicting relative patch positions, extended by jigsaw (Noroozi & Favaro, 2016) and content restoration (Kim et al., 2018). DeepPermNet (Santa Cruz et al., 2017) learns permutation structures, MP3 (Zhai et al., 2022) predicts absolute patch locations, and LOCA (Caron et al., 2024) predicts relative positions of clustered patches. In NLP, StructBERT (Wang et al., 2019), ALBERT (Lan et al., 2019), and SLM (Lee et al., 2020) use sentence order prediction and sequence restoration, while Nandy *et al.* (Nandy et al., 2024) extend to permutation-based objectives. Beyond vision and language, GeoMAE (Tian et al., 2023) reconstructs geometric features of masked point clouds, and LEGO (Sun et al., 2024) recovers perturbed molecular geometries. While conceptually related, these methods operate on discrete, domain-specific positional targets (e.g., patch indices, sentence order). By contrast, our approach reconstructs continuous physical positions of sensor nodes, naturally aligning with distributed sensing.

## 3 METHOD

To develop a pretraining method that effectively utilizes the unique placement characteristics of sensing nodes, we propose SPAR, which explicitly leverages **the duality between observer placement and signals** in the distributed sensing data. Specifically, we extend the traditional MAE framework by introducing **spatial positional embeddings** to represent device locations, and **structural positional embeddings** to encapsulate effects of other placement characteristics (such as orientation). Furthermore, we propose to optimize our model with novel **dual reconstruction objectives** to enhance its ability to retain both signal and spatial information in its learned representations. The overall architecture of SPAR is shown in Figure 2, with each component detailed below. Importantly, SPAR is grounded in solid theoretical foundations from both information theory and the study of occlusion-invariant representations, offering deep insights into our design.

For clarity, we adopt the following notation convention throughout the paper: scalars are denoted by lowercase or uppercase letters (e.g., $k, K$), matrices by bold uppercase letters (e.g., $\boldsymbol{R}, \boldsymbol{S}$), tensors by bold calligraphic letters (e.g., $\boldsymbol{\mathcal{R}}, \boldsymbol{\mathcal{S}}$), and random tensor variables by sans-serif uppercase letters (e.g., $\mathsf{R}, \mathsf{S}$). We use $\mathcal{F}$ with appropriate subscripts to denote the forward operations of various transformer-based modules. A summary of notations is provided in Table 9 in Appendix C.

### 3.1 EMBEDDING FOR SIGNALS, SPATIAL POSITIONS, AND STRUCTURAL POSITIONS

We consider a multi-modality distributed sensing system with $K$ modalities, where the $k$-th modality ($k \in \{1, \ldots, K\}$) consists of $n^{(k)}$ sensing nodes. The signals collected from these nodes are first

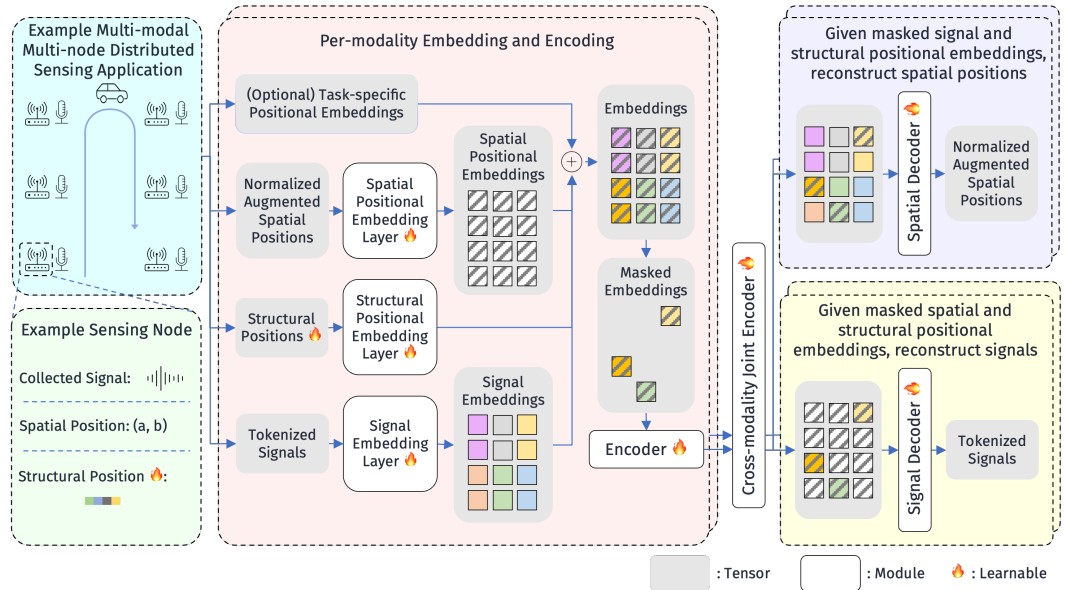

Figure 2: Architectural overview of SPAR. Each node is assigned a continuous learnable structural position to capture its unique characteristics. The signals, spatial positions, and structural positions of all nodes are projected into a shared embedding space, combined, and encoded into latent embeddings. The latent embeddings are optimized with dual reconstruction objectives, encouraging the model to effectively utilize and retain both signal and positional information in a self-supervised and context-aware manner.

tokenized to be compatible with transformer encoders. The tokenization strategy is task-specific and flexible—for example, IMU time-series data can be divided into temporal segments, while acoustic spectrograms can be split into patches. We denote the tokenized signals as $\boldsymbol{\mathcal{X}}^{(k)} \in \mathbb{R}^{n^{(k)} \times m^{(k)} \times d_{\boldsymbol{\mathcal{X}}}^{(k)}}$, where $m^{(k)}$ is the number of tokens and $d_{\boldsymbol{\mathcal{X}}}^{(k)}$ is the token dimension. We then project these tokens into the transformer embedding space using a learnable linear layer, as $\widetilde{\boldsymbol{\mathcal{X}}}_{i,j,:}^{(k)} = \mathcal{F}_{\text{sig\_embed}}^{(k)}(\boldsymbol{\mathcal{X}}_{i,j,:}^{(k)})$, yielding signal embeddings $\widetilde{\boldsymbol{\mathcal{X}}}^{(k)} \in \mathbb{R}^{n^{(k)} \times m^{(k)} \times d}$, where $d$ is the transformer model dimension.

A distinguishing aspect of distributed sensing data is the availability of **spatial positions** of the nodes, reflecting their physical layout in the field, which can be denoted as $\boldsymbol{S}^{(k)} \in \mathbb{R}^{n^{(k)} \times d_{\boldsymbol{S}}}$. For instance, in Figure 2, the spatial positions are two-dimensional, indicating longitudinal and lateral node locations. Unlike the discrete ordinal indices typically used in NLP (Vaswani et al., 2017) or CV (Dosovitskiy et al., 2020), spatial positions in distributed sensing data are continuous vectors, making classical positional embedding strategies unsuitable (Vaswani et al., 2017; Dosovitskiy et al., 2020; Su et al., 2024; Press et al., 2021). To address this, we propose to continuously project the spatial positions into the embedding space as $\widetilde{\boldsymbol{S}}_{i,j,:}^{(k)} = \mathcal{F}_{\text{sp\_embed}}^{(k)}(\boldsymbol{\mathcal{S}}_{i,j,:}^{(k)})$, where $\boldsymbol{\mathcal{S}}_{:,j,:}^{(k)} = \boldsymbol{S}^{(k)}$ is the spatial positions broadcasted to match the dimension of the tokenized signals. The spatial positional embeddings $\widetilde{\boldsymbol{S}}^{(k)}$ are then added to the signal embeddings to incorporate spatial context into the model.

However, two challenges arise in practice. First, spatial positions may vary widely in absolute locations and scales. For the example in Figure 2, data may be collected in different cities, with some layouts covering small parking lots and others spanning large open areas. To ensure consistency, we **normalize** the spatial positions of each sample to have zero mean and unit variance. Second, existing datasets often contain only a limited number of distinct spatial layouts for which data were collected, leading the spatial embeddings (and the model) to overfit in pretraining, reducing generalizability to potentially unseen spatial arrangements during fine-tuning or testing. To mitigate this, we apply **geometric augmentation** during pretraining by randomly rotating and translating the normalized spatial positions, improving robustness to unseen layouts.

While spatial positions capture physical layout, they do not fully represent structural placement conditions, such as the body part a sensor is attached to, or the orientation used for a directional

measurement device (e.g., front-facing versus rear-facing camera on an autonomous car). Manually labeling these characteristics for all nodes is often costly and non-scalable. To address this, we assign each node a continuous learnable vector, called **structural position**. The structural positions for all nodes are denoted as $\boldsymbol{R}^{(k)} \in \mathbb{R}^{n^{(k)} \times d_{\boldsymbol{R}}}$, where we typically choose the dimension of structural position $d_{\boldsymbol{R}} \ll d$ to ensure training efficiency and scalability to large-scale sensing applications. As with spatial positions, we broadcast $\boldsymbol{R}^{(k)}$ to form $\boldsymbol{\mathcal{R}}^{(k)} \in \mathbb{R}^{n^{(k)} \times m^{(k)} \times d_{\boldsymbol{R}}^{(k)}}$, project it into the embedding space via $\widetilde{\boldsymbol{\mathcal{R}}}_{i,j,:}^{(k)} = \mathcal{F}_{\text{st\_embed}}^{(k)}(\boldsymbol{\mathcal{R}}_{i,j,:}^{(k)})$, and add it to the signal embeddings. These learnable structural positions are trained jointly with the rest of the model in the pretraining stage, enabling it to automatically capture node-specific information.

Beyond this learnable formulation, we further explore leveraging **Large Language Models (LLMs)** to derive structural positional embeddings from free-form textual metadata describing each sensor's placement, modality, and signal characteristics. These LLM-derived embeddings, obtained by encoding textual descriptions, are projected and fused with the other embeddings but kept frozen during pretraining. This metadata-driven variant, which we denote as **SPAR+LLM**, encourages generalization to previously unseen sensors and placements.

Structural positions bear an interesting mathematical interpretation: if we assume the influence of a node's unique characteristics on its signal embedding can be summarized as an additive vector, which lies within a specific subspace of the embedding space, then the weight matrix of $\mathcal{F}_{\text{st\_embed}}^{(k)}$ can be understood as a learned set of basis vectors spanning this subspace. The structural position of each node can then be viewed as the coordinate of the corresponding additive vector in this subspace, thereby substantiating its meaning as an abstract "position".

## 3.2 Masked Autoencoding with Dual Reconstruction Objectives

After combining the signal embeddings $\widetilde{\boldsymbol{\mathcal{X}}}^{(k)}$, spatial positional embeddings $\widetilde{\boldsymbol{\mathcal{S}}}^{(k)}$, and structural positional embeddings $\widetilde{\boldsymbol{\mathcal{R}}}^{(k)}$ (as well as any additional task-specific positional embeddings, such as 2D patch positions in a spectrogram, which we omit in the rest of this paper for clarity), we apply a binary mask $\boldsymbol{M}^{(k)} \in \{0,1\}^{n^{(k)} \times m^{(k)}}$ over the combined embeddings to randomly mask out a fraction of tokens. The unmasked tokens are then fed into a per-modality transformer encoder to produce latent embeddings $\boldsymbol{Z}^{(k)}$:

$$\boldsymbol{Z}^{(k)} = \mathcal{F}_{\text{enc}}^{(k)}(\texttt{mask}(\widetilde{\boldsymbol{\mathcal{X}}}^{(k)} + \widetilde{\boldsymbol{\mathcal{S}}}^{(k)} + \widetilde{\boldsymbol{\mathcal{R}}}^{(k)}; \boldsymbol{M}^{(k)})), \tag{1}$$

where $\texttt{mask}(\cdot; \cdot)$ denotes the masking operation. To enable cross-modal interactions, we then apply a joint transformer encoder over the concatenated latent embeddings from all modalities:

$$(\widetilde{\boldsymbol{Z}}^{(1)}, \ldots, \widetilde{\boldsymbol{Z}}^{(K)}) = \mathcal{F}_{\text{joint\_enc}}(\texttt{concat}(\boldsymbol{Z}^{(1)}, \ldots, \boldsymbol{Z}^{(K)})), \tag{2}$$

where $\texttt{concat}(\cdot)$ denotes contatenation, and $\widetilde{\boldsymbol{Z}}^{(k)}$ denotes the post-fusion latent embeddings for the $k$-th modality. In the fine-tuning stage, the encoders are frozen, and the post-fusion latent embeddings are extracted and passed into a task-specific prediction head, which is trained using appropriate supervised objectives.

During the pretraining stage, however, the post-fusion latent embeddings are decoded, enabling the encoders to be optimized with self-supervised objectives. In the standard MAE framework, a single decoder is typically used to reconstruct the masked signals, which overlooks the rich spatial and structural context inherent in distributed sensing data. To address this, we introduce two decoders with **dual reconstruction objectives**, explicitly exploiting the duality between positions and signals. Specifically, the **signal decoder** is tasked with reconstructing the masked signals, using both the latent embeddings and the masked spatial and structural positional embeddings:

$$\widehat{\boldsymbol{\mathcal{X}}}^{(k)} = \mathcal{F}_{\text{sig\_dec}}^{(k)}(\texttt{concat}(\widetilde{\boldsymbol{Z}}^{(k)}, \texttt{mask}(\widetilde{\boldsymbol{\mathcal{S}}}^{(k)} + \widetilde{\boldsymbol{\mathcal{R}}}^{(k)}; \overline{\boldsymbol{M}}^{(k)}))), \tag{3}$$

where $\overline{\boldsymbol{M}}^{(k)} = 1 - \boldsymbol{M}^{(k)}$ is the complement mask, and $\widehat{\boldsymbol{\mathcal{X}}}^{(k)}$ denotes the reconstructed signals. In parallel, the **spatial decoder** is responsible for reconstructing the masked spatial positions, conditioned on the latent embeddings and the masked signal and structural positional embeddings:

$$\widehat{\boldsymbol{\mathcal{S}}}^{(k)} = \mathcal{F}_{\text{sp\_dec}}^{(k)}(\texttt{concat}(\widetilde{\boldsymbol{Z}}^{(k)}, \texttt{mask}(\widetilde{\boldsymbol{\mathcal{X}}}^{(k)} + \widetilde{\boldsymbol{\mathcal{R}}}^{(k)}; \overline{\boldsymbol{M}}^{(k)}))), \tag{4}$$

where $\widehat{\boldsymbol{S}}^{(k)}$ denotes the reconstructed spatial positions. The loss $L$ used to train our model combines the Mean Squared Error (MSE) reconstruction losses over both decoders:

$$L = \sum_{k=1}^{K} \|\mathrm{mask}(\boldsymbol{\mathcal{X}}^{(k)} - \widehat{\boldsymbol{\mathcal{X}}}^{(k)}; \overline{\boldsymbol{M}}^{(k)})\|_2^2 + \|\mathrm{mask}(\boldsymbol{S}^{(k)} - \widehat{\boldsymbol{S}}^{(k)}; \overline{\boldsymbol{M}}^{(k)})\|_2^2. \tag{5}$$

Our dual reconstruction objectives compel the model to extract, utilize, and preserve the full spectrum of signal, spatial, and structural information.

A practical challenge in multi-modal, multi-node distributed sensing systems is the frequent occurrence of missing data due to hardware failures or unreliable communication links. To mitigate their impact, we pad missing entries with zeros and exclude them from the reconstruction loss by setting their corresponding loss terms to zero.

### 3.3 THEORETICAL ANALYSES

In this subsection, we provide theoretical support for the design of SPAR, drawing from principles in both information theory and occlusion-invariant representation learning. These insights help illuminate the rationale behind SPAR's design.

**Analysis from the Perspective of Information Theory.** We first analyze SPAR in comparison to classical MAE through the lens of information theory, as formalized in the following proposition:

**Proposition 3.1.** *Let $\mathsf{X}^{(k)}, \widetilde{\mathsf{Z}}^{(k)}, \mathsf{S}^{(k)}, \mathsf{R}^{(k)}$ denote the random variables corresponding to the signals, the post-fusion latent embeddings, the spatial positions, and the structural positions, for $k \in \{1, \ldots, K\}$, respectively. Let $\mathbb{E}[L']$ and $\mathbb{E}[L]$ denote the expected losses of classical MAE and SPAR over the data distribution, respectively, and let $C'$ and $C$ be constants independent of model parameters. Then, under certain assumptions, for classical MAE, we can have the following bound:*

$$-\mathbb{E}[L'] + C' \le \sum_{k=1}^{K} I(\mathsf{X}^{(k)}; \widetilde{\mathsf{Z}}^{(k)}), \tag{6}$$

*where $I(\cdot; \cdot)$ denotes mutual information. In contrast, for SPAR, we can have*

$$-\mathbb{E}[L] + C \le \sum_{k=1}^{K} I(\mathsf{X}^{(k)}; \widetilde{\mathsf{Z}}^{(k)} | \mathsf{S}^{(k)}, \mathsf{R}^{(k)}) + I(\mathsf{S}^{(k)}; \widetilde{\mathsf{Z}}^{(k)} | \mathsf{X}^{(k)}, \mathsf{R}^{(k)}). \tag{7}$$

*where $I(\cdot; \cdot | \cdot)$ denotes conditional mutual information.*

The proof is detailed in Appendix D.1. This result highlights a key distinction between classical MAE and SPAR. In classical MAE, minimizing the expected loss encourages latent embeddings to retain information about the input signals, but without explicitly incorporating spatial or structural context. In contrast, SPAR is designed to promote embeddings that capture signal information beyond what is explained by structural and spatial cues, and similarly, to retain spatial information conditioned on the signal and structural characteristics. This encourages the embeddings to be context-aware and jointly informative of both signals and spatial layout, while avoiding memorizing redundant information.

**Analysis from the Perspective of Occlusion-invariant Representation.** We next analyze SPAR through the lens of occlusion-invariant representation learning. For clarity and readability, we present the analysis for a single modality by omitting the superscript $(k)$; the generalization to the multi-modality case is straightforward. The core result is formalized in the following proposition:

**Proposition 3.2.** *As shown by Kong et al. (Kong & Zhang, 2023), classical MAE can be viewed as a form of contrastive learning, where the positive pair consists of two complementary masked views of the signals:*

$$\left[\mathrm{mask}(\boldsymbol{\mathcal{X}}; \boldsymbol{M}), \quad \mathrm{mask}(\boldsymbol{\mathcal{X}}; \overline{\boldsymbol{M}})\right]. \tag{8}$$

*In contrast, SPAR can be interpreted as performing contrastive learning over two types of enriched positive pairs: 1) complementary masked views of signals with shared spatial and structural context:*

$$\left[(\mathrm{mask}(\boldsymbol{\mathcal{X}}; \boldsymbol{M}), \boldsymbol{S}, \boldsymbol{\mathcal{R}}), \quad (\mathrm{mask}(\boldsymbol{\mathcal{X}}; \overline{\boldsymbol{M}}), \boldsymbol{S}, \boldsymbol{\mathcal{R}})\right], \tag{9}$$

*and 2) complementary masked views of spatial positions with shared signal and structural context:*

$$\left[(\boldsymbol{\mathcal{X}}, \mathrm{mask}(\boldsymbol{S}; \boldsymbol{M}), \boldsymbol{\mathcal{R}}), \quad (\boldsymbol{\mathcal{X}}, \mathrm{mask}(\boldsymbol{S}; \overline{\boldsymbol{M}}), \boldsymbol{\mathcal{R}})\right]. \tag{10}$$

Table 1: Comparison of the MSE and averaged Distance Error between SPAR and baselines on M3N-VC single-vehicle localization task. The label ratio during fine-tuning varies from 1.0 to 0.2.

| Method | M3N-VC Single-vehicle Localization | | | | | |
| | Label Ratio 1.0 | | Label Ratio 0.5 | | Label Ratio 0.2 | |
| | MSE $(m^2)$ $(\downarrow)$ | Dist. Err. $(m)$ $(\downarrow)$ | MSE $(m^2)$ $(\downarrow)$ | Dist. Err. $(m)$ $(\downarrow)$ | MSE $(m^2)$ $(\downarrow)$ | Dist. Err. $(m)$ $(\downarrow)$ |
|---|---|---|---|---|---|---|
| CMC | $51.11 \pm 14.67$ | $6.76 \pm 0.75$ | $71.81 \pm 15.32$ | $7.99 \pm 0.64$ | $111.37 \pm 8.02$ | $11.05 \pm 0.57$ |
| Cosmo | $38.40 \pm 4.14$ | $6.03 \pm 0.21$ | $53.12 \pm 9.75$ | $7.19 \pm 0.40$ | $97.08 \pm 9.49$ | $10.95 \pm 0.57$ |
| SimCLR | $34.40 \pm 4.47$ | $5.64 \pm 0.25$ | $45.14 \pm 7.34$ | $6.57 \pm 0.08$ | $74.53 \pm 3.13$ | $9.48 \pm 0.17$ |
| AudioMAE | $22.36 \pm 0.49$ | $5.40 \pm 0.11$ | $30.12 \pm 2.97$ | $6.33 \pm 0.28$ | $41.75 \pm 3.30$ | $7.47 \pm 0.28$ |
| CAV-MAE | $18.85 \pm 0.41$ | $5.06 \pm 0.04$ | $22.90 \pm 0.82$ | $5.58 \pm 0.12$ | $24.84 \pm 0.33$ | $5.78 \pm 0.10$ |
| FOCAL | $32.43 \pm 4.68$ | $5.37 \pm 0.22$ | $40.84 \pm 2.82$ | $6.20 \pm 0.19$ | $69.62 \pm 5.62$ | $8.50 \pm 0.35$ |
| FreqMAE | $29.61 \pm 2.85$ | $5.36 \pm 0.16$ | $42.06 \pm 14.44$ | $6.25 \pm 0.70$ | $91.40 \pm 35.32$ | $9.15 \pm 1.27$ |
| PhyMask | $28.02 \pm 5.91$ | $5.29 \pm 0.33$ | $33.74 \pm 2.18$ | $5.85 \pm 0.12$ | $64.36 \pm 4.70$ | $8.44 \pm 0.36$ |
| SPAR | $\mathbf{12.98 \pm 0.11}$ | $\mathbf{4.20 \pm 0.07}$ | $\mathbf{15.07 \pm 1.03}$ | $\mathbf{4.51 \pm 0.09}$ | $\mathbf{21.36 \pm 0.62}$ | $\mathbf{5.40 \pm 0.04}$ |

The proof is detailed in Appendix D.2. This formulation highlights another key distinction: by treating masked views of the signal embeddings as positive pairs, classical MAE promotes occlusion-invariant representations solely within the signal domain, without accounting for spatial or structural positions. In contrast, SPAR encourages representations to be invariant to occlusion in both the signal and spatial domains, while preserving the presence of each other and the structural characteristics, leading to more robust and context-aware learned representations.

## 4 EVALUATION

In this section, we present our experimental evaluation of SPAR on three multiple multi-modal, multi-node distributed sensing datasets spanning diverse sensing modalities and spatial scales. To ensure a fair comparison, all baseline methods and our model use the same ViT backbone architecture (Dosovitskiy et al., 2020) and identical task-specific prediction heads. Pretraining and fine-tuning are conducted for the same number of epochs across all methods. All reported results are aggregated over three random seeds. The prediction heads are designed to be lightweight and straightforward, tailored to the needs of each downstream task. Detailed descriptions of each task setup can be found in Appendix E.

**Datasets.** We conducted experiments on three real-world distributed sensing datasets: (1) the M3N-VC dataset(Li et al., 2025), which includes acoustic and seismic signals from moving vehicles, collected across six distinct outdoor scenes; (2) the Ridgecrest Seismicity Dataset(California Institute of Technology (Caltech), 1926), containing multi-modal seismic waveform recordings of earthquake events in the Ridgecrest region of California; and (3) the RealWorld-HAR dataset(Sztyler & Stuckenschmidt, 2016), comprising accelerometer, gyroscope, and magnetometer readings for human activity recognition. Further dataset details are available in Appendix E.

**Baselines.** We compare SPAR against eight state-of-the-art baseline methods: CMC (Tian et al., 2020), Cosmo (Ouyang et al., 2022), SimCLR (Chen et al., 2020), AudioMAE (Huang et al., 2022), CAV-MAE (Gong et al., 2022), FOCAL (Liu et al., 2023), FreqMAE (Kara et al., 2024b), and PhyMask (Kara et al., 2024a). Among these, CMC, Cosmo, SimCLR, and FOCAL are contrastive learning-based methods, while AudioMAE, CAV-MAE, and FreqMAE follow the masked autoencoding (MAE) paradigm. Please see Appendix E.1 for a detailed description of each baseline.

### 4.1 EVALUATION ON M3N-VC DATASET

We begin with the M3N-VC dataset, focusing on the task of **single-vehicle localization**, where the goal is to predict the position of a vehicle within the monitored area. We pretrain on the full dataset and finetune only the prediction head (a single transformer layer) on scene "H24," which contains a single moving vehicle. To test robustness under limited supervision, we vary the ratio of labeled data from 100% to 20%. As shown in Table 1, SPAR consistently achieves the lowest MSE and Distance Error across all label ratios, demonstrating resilience to scarce supervision. Example localization visualizations are provided in Figure 3a (Appendix B).

Table 2: Comparison of the mAP@$r$ metric ($r$ is the distance threshold varying across {2,4,6,8} meters) between SPAR and baselines on M3N-VC multi-vehicle joint classification and localization task.

| Method | M3N-VC Multi-vehicle Joint Classification and Localization | | | |
|---|---|---|---|---|
| | mAP@4m (%) (↑) | mAP@6m (%) (↑) | mAP@8m (%) (↑) | mAP@10m (%) (↑) |
| CMC | $0.06 \pm 0.05$ | $0.48 \pm 0.36$ | $1.61 \pm 1.10$ | $3.62 \pm 2.19$ |
| Cosmo | $0.16 \pm 0.05$ | $1.66 \pm 0.23$ | $4.77 \pm 0.72$ | $9.52 \pm 1.20$ |
| SimCLR | $0.31 \pm 0.14$ | $2.22 \pm 0.58$ | $6.53 \pm 1.24$ | $13.07 \pm 2.08$ |
| AudioMAE | $1.39 \pm 0.48$ | $6.96 \pm 1.42$ | $17.11 \pm 3.24$ | $28.98 \pm 4.01$ |
| CAV-MAE | $22.12 \pm 2.94$ | $52.08 \pm 4.16$ | $73.41 \pm 3.24$ | $85.36 \pm 1.78$ |
| FOCAL | $0.08 \pm 0.05$ | $0.82 \pm 0.40$ | $2.94 \pm 1.04$ | $6.82 \pm 1.99$ |
| FreqMAE | $0.24 \pm 0.01$ | $1.67 \pm 0.32$ | $5.34 \pm 0.99$ | $11.31 \pm 1.49$ |
| PhyMask | $0.08 \pm 0.03$ | $0.88 \pm 0.24$ | $3.04 \pm 0.74$ | $6.64 \pm 1.46$ |
| SPAR | $\mathbf{41.57 \pm 2.69}$ | $\mathbf{71.82 \pm 3.69}$ | $\mathbf{86.28 \pm 1.77}$ | $\mathbf{92.99 \pm 0.79}$ |

Table 3: Comparison between SPAR and baselines across three tasks: (1) M3N-VC single-vehicle classification, (2) Ridgecrest Seismicity Dataset earthquake localization, and (3) RealWorld-HAR activity recognition. Each block reports task-specific metrics.

| Method | M3N-VC Classification | | Ridgecrest Earthquake Localization | | RealWorld-HAR Recognition | |
|---|---|---|---|---|---|---|
| | Accuracy (%) (↑) | F1 (%) (↑) | MSE ($km^2$) (↓) | Dist. Err. ($km$) (↓) | Accuracy (%) (↑) | F1 (%) (↑) |
| CMC | $89.53 \pm 7.62$ | $89.33 \pm 7.78$ | $94.25 \pm 6.67$ | $10.38 \pm 0.63$ | $74.97 \pm 1.23$ | $74.82 \pm 2.18$ |
| Cosmo | $94.21 \pm 0.50$ | $94.04 \pm 0.54$ | $98.24 \pm 13.77$ | $10.44 \pm 0.83$ | $84.37 \pm 0.33$ | $85.30 \pm 0.43$ |
| SimCLR | $95.53 \pm 0.73$ | $95.41 \pm 0.74$ | $99.87 \pm 11.31$ | $10.29 \pm 0.52$ | $84.36 \pm 0.47$ | $85.49 \pm 0.36$ |
| AudioMAE | $99.06 \pm 0.23$ | $99.03 \pm 0.24$ | $33.65 \pm 3.51$ | $5.65 \pm 0.29$ | $89.18 \pm 0.32$ | $90.11 \pm 0.53$ |
| CAV-MAE | $98.97 \pm 0.04$ | $98.94 \pm 0.04$ | $31.58 \pm 3.57$ | $5.48 \pm 0.37$ | $88.12 \pm 0.24$ | $89.05 \pm 0.35$ |
| FOCAL | $93.62 \pm 0.75$ | $93.46 \pm 0.76$ | $131.50 \pm 1.48$ | $12.53 \pm 0.09$ | $84.98 \pm 0.73$ | $86.24 \pm 0.77$ |
| FreqMAE | $92.72 \pm 0.75$ | $92.55 \pm 0.79$ | $54.08 \pm 5.44$ | $7.14 \pm 0.25$ | $83.43 \pm 0.56$ | $84.07 \pm 0.50$ |
| PhyMask | $83.38 \pm 2.33$ | $82.68 \pm 2.27$ | $56.39 \pm 3.27$ | $7.67 \pm 0.39$ | $84.79 \pm 3.23$ | $82.15 \pm 9.13$ |
| SPAR | $\mathbf{99.27 \pm 0.07}$ | $\mathbf{99.26 \pm 0.07}$ | $\mathbf{23.46 \pm 2.77}$ | $\mathbf{5.37 \pm 0.24}$ | $\mathbf{89.63 \pm 0.57}$ | $\mathbf{90.45 \pm 0.63}$ |

The second task is **single-vehicle classification**, where the model distinguishes among four vehicle types and a background class. The setup mirrors that of localization. As reported in Table 3, SPAR attains the highest accuracy and F1 score among all methods. The confusion matrix (Figure 5) and T-SNE plot (Figure 4) confirm that the learned embeddings cleanly separate all five classes, underscoring their discriminative power.

We next consider the more challenging task of **multi-vehicle joint classification and localization**, where multiple vehicles move simultaneously, generating overlapping signals. We pretrain on the full dataset and finetune on scene "I22," which includes multiple vehicles. A 2-layer transformer head with a DETR-style loss (Carion et al., 2020) is used. To evaluate performance, we adopt mAP@$r$ from object detection (Lin et al., 2014), where predictions are correct only if both class and location are accurate within radius $r$. As shown in Table 2, SPAR significantly outperforms all baselines across thresholds, including strict ones (e.g., 4m), despite noisy 1Hz smartphone GPS labels. This highlights its strong spatial reasoning under complex conditions. Additional visualizations of predictions are included in Figure 6 (Appendix B).

Finally, we conduct three complementary evaluations (details in Appendix A): (1) **Lossy Communication**: SPAR remains robust under random node-level data dropout, outperforming baselines (Table 6). (2) **Unseen Sensor Placements**: The pretrained, frozen encoders of SPAR generalize well to unseen placement, confirming placement-aware robustness (Table 7). (3) **Ablations**: The ablation studies indicate that each design component in SPAR contributes meaningfully and synergically, and SPAR maintains robust to varing hyperparameters (Table 8).

### 4.2 EVALUATION ON RIDGECREST SEISMICITY DATASET

We next evaluate SPAR on the Ridgecrest Seismicity Dataset for **earthquake event localization**. Here, the goal is to predict 3D earthquake coordinates from multi-modal seismic waveforms collected across 16 monitoring stations. Compared to vehicle monitoring, this task involves a much larger

Table 5: Impact of data compression on SPAR across three tasks: (1) M3N-VC single-vehicle classification, (2) Ridgecrest Seismicity Dataset earthquake localization, and (3) RealWorld-HAR activity recognition.

| Method | M3N-VC Single-vehicle Localization | | | Ridgecrest Earthquake Localization | | | RealWorld-HAR Recognition | | |
|---|---|---|---|---|---|---|---|---|---|
| | Traffic (%) ↓ | MSE ↓ | Dist. Err. ↓ | Traffic (%) ↓ | MSE ↓ | Dist. Err. ↓ | Traffic (%) ↓ | Acc. ↑ | F1 ↑ |
| SPAR | 100.00 | 12.12 | 4.12 | 100.00 | 22.17 | 5.10 | 100.00 | 90.23 | 91.11 |
| SPAR w. Compression | **10.70** | 12.26 | 4.13 | **6.30** | 22.18 | 5.10 | **24.03** | 90.00 | 90.91 |

spatial scale (tens of kilometers) and a 20.38% inherent missing-data rate, as weak seismic waves often fail to reach distant stations. Despite these challenges, SPAR achieves the lowest MSE and Distance Error among all methods (Table 3), demonstrating strong spatial reasoning in large-scale, partially observed environments. Visualizations of predictions are provided in Figure 3b (Appendix B).

## 4.3 EVALUATION ON REALWORLD-HAR DATASET

Table 4: Comparison of the Accuracy and F1 score between SPAR and SPAR+LLM on the RealWorld-HAR human activity recognition task. An example text metadata is "Sensor 0: a smartphone-mounted accelerometer on the Head, capturing low-amplitude time-series signals along the x, y, and z axes reflecting subtle head motions and posture shifts."

| Method | RealWorld-HAR Recognition | |
|---|---|---|
| | Accuracy (%) (↑) | F1 (%) (↑) |
| SPAR | $89.63 \pm 0.57$ | $90.45 \pm 0.63$ |
| SPAR+LLM | $\mathbf{90.40 \pm 0.68}$ | $\mathbf{91.13 \pm 0.59}$ |

Finally, we evaluate SPAR on the RealWorld-HAR dataset for **human activity recognition** using IMU signals. This task differs from the above in operating at a smaller spatial scale but involving highly diverse placements: sensors mounted on the wrist, ankle, chest, etc., yield signals with distinct characteristics. Despite this heterogeneity, SPAR achieves the best accuracy and F1 among all baselines (Table 3), underscoring its robustness to placement diversity. The confusion matrix (Figure 5) and t-SNE visualizations (Figure 4) show that predicted activity patterns align well with the conceptual separability of classes.

We also evaluate **SPAR+LLM**, elaborated in Section 3.1, which replaces learnable structural positions with LLM-derived embeddings from textual sensor metadata. On this dataset, where we can create rich natural language descriptions of sensor placement, **SPAR+LLM** yields additional gains over SPAR (Table 4). This highlights the benefit of combining spatial priors with semantic context for placement-aware learning.

## 4.4 ROBUSTNESS UNDER COMMUNICATION CONSTRAINTS

Distributed sensing often operates under limited bandwidth, restricting the transmission of raw sensor data. To test SPAR in such settings, we compress raw inputs using standard formats, such as JPEG for M3N-VC and Ridgecrest, and WebP for RealWorld-HAR. As shown in Table 5, compression reduces communication traffic by up to 90% with negligible performance loss across all tasks. This demonstrates that SPAR is robust to severe bandwidth constraints, making it practical for real-world deployments.

## 5 CONCLUSION

This paper presents SPAR, a general self-supervised pretraining framework designed for the whole spectrum of multi-modal, multi-node distributed sensing. By introducing spatial and structural positional embeddings alongside dual reconstruction objectives, SPAR leverages the inherent duality between observer positions and observed signals to enable placement-aware representation learning. Theoretical analyses grounded in information theory and occlusion-invariant learning offer principled support for the framework. Extensive experiments across three real-world datasets, spanning diverse sensing modalities, placement configurations, and task types, demonstrate the superior generalizability and robustness of SPAR. We hope SPAR inspires broader efforts toward integrating spatial and structural context into foundational representation learning paradigms for the wide range of distributed sensing applications.

ETHICS STATEMENT

This work develops a general pretraining framework for distributed sensing using only publicly available datasets that contain no personally identifiable or sensitive information. Our study does not involve human subjects or private data collection, and we believe it poses no direct ethical concerns.

REPRODUCIBILITY STATEMENT

We have taken concrete steps to ensure the reproducibility of our results. Detailed descriptions of datasets, preprocessing procedures, and training protocols are provided in Appendix E. Formal proofs of our theoretical results are included in Appendix D. In addition, we release our implementation and scripts at `https://anonymous.4open.science/r/SPAR-4C74/`.

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

# APPENDIX

## A    ADDITIONAL EXPERIMENTS RESULTS

Beyond the primary experiments in Section 4.1, we conduct additional studies to further evaluate the robustness and generalization of SPAR.

**Lossy Communication.** We first examine SPAR's resilience to missing data caused by message drops, a common challenge in real-world sensor networks. Each node's data is independently dropped with probabilities of 5%, 10%, or 20%. As shown in Table 6, SPAR consistently achieves the lowest mean squared error and distance error across all settings, demonstrating strong localization performance even under substantial input loss.

Table 6: Comparison of the MSE and Distance Error between SPAR and baselines on M3N-VC single-vehicle localization task, under various message drop rates.

| Method | M3N-VC Single-vehicle Localization | | | | | |
| | Message Drop Rate 0.05 | | Message Drop Rate 0.1 | | Message Drop Rate 0.2 | |
| | MSE $(m^2)$ $(\downarrow)$ | Dist. Err. $(m)$ $(\downarrow)$ | MSE $(m^2)$ $(\downarrow)$ | Dist. Err. $(m)$ $(\downarrow)$ | MSE $(m^2)$ $(\downarrow)$ | Dist. Err. $(m)$ $(\downarrow)$ |
|---|---|---|---|---|---|---|
| CMC | $50.13 \pm 17.69$ | $6.48 \pm 0.91$ | $50.27 \pm 18.03$ | $6.60 \pm 1.17$ | $55.38 \pm 22.01$ | $6.91 \pm 1.29$ |
| Cosmo | $29.97 \pm 2.30$ | $5.44 \pm 0.12$ | $29.37 \pm 2.43$ | $5.44 \pm 0.10$ | $33.19 \pm 2.67$ | $5.61 \pm 0.12$ |
| SimCLR | $26.77 \pm 2.55$ | $5.16 \pm 0.07$ | $25.65 \pm 1.28$ | $5.15 \pm 0.07$ | $28.31 \pm 3.94$ | $5.4 \pm 0.18$ |
| AudioMAE | $19.29 \pm 1.42$ | $4.91 \pm 0.17$ | $18.25 \pm 1.23$ | $4.77 \pm 0.10$ | $19.03 \pm 1.14$ | $4.77 \pm 0.05$ |
| CAV-MAE | $16.28 \pm 0.17$ | $4.68 \pm 0.03$ | $15.85 \pm 0.81$ | $4.57 \pm 0.10$ | $15.98 \pm 1.67$ | $4.44 \pm 0.12$ |
| FOCAL | $26.62 \pm 1.02$ | $5.21 \pm 0.17$ | $27.42 \pm 2.33$ | $5.32 \pm 0.20$ | $33.03 \pm 2.28$ | $5.70 \pm 0.15$ |
| FreqMAE | $27.48 \pm 1.43$ | $5.14 \pm 0.11$ | $28.32 \pm 1.22$ | $5.17 \pm 0.14$ | $28.68 \pm 6.96$ | $5.32 \pm 0.24$ |
| PhyMask | $23.18 \pm 4.96$ | $4.98 \pm 0.32$ | $24.09 \pm 3.31$ | $5.03 \pm 0.27$ | $27.37 \pm 2.04$ | $5.31 \pm 0.21$ |
| SPAR | $\mathbf{12.65 \pm 0.61}$ | $\mathbf{4.09 \pm 0.11}$ | $\mathbf{12.48 \pm 0.68}$ | $\mathbf{4.07 \pm 0.02}$ | $\mathbf{14.56 \pm 2.47}$ | $\mathbf{4.27 \pm 0.07}$ |

**Unseen Sensor Placements**. Next, we assess generalization to unseen sensor placements. All models are pretrained on the full dataset excluding scenes H08 and H24 (which share similar configurations), then finetuned and evaluated on H24. To simulate transfer, nodes in H24 are assigned structural position vectors randomly drawn from those learned during pretraining. As reported in Table 7, SPAR continues to outperform baselines, underscoring its placement-aware generalization ability.

Table 7: Comparison of the MSE and Distance Error between SPAR and baselines on M3N-VC single-vehicle localization task. SPAR and baselines are finetuned and evaluated on a placement unseen in the pretraining.

| Method | M3N-VC Single-vehicle Localization (Finetuned and Evaluated on Unseen Placement) | |
| | MSE $(m^2)$ $(\downarrow)$ | Dist. Err. $(m)$ $(\downarrow)$ |
|---|---|---|
| CMC | $61.78 \pm 17.68$ | $7.23 \pm 0.78$ |
| Cosmo | $63.43 \pm 12.85$ | $7.22 \pm 0.63$ |
| SimCLR | $35.82 \pm 7.57$ | $5.92 \pm 0.39$ |
| AudioMAE | $41.25 \pm 4.60$ | $6.64 \pm 0.31$ |
| CAV-MAE | $37.01 \pm 0.68$ | $6.27 \pm 0.04$ |
| FOCAL | $41.79 \pm 11.04$ | $5.91 \pm 0.41$ |
| FreqMAE | $30.65 \pm 1.14$ | $5.51 \pm 0.19$ |
| PhyMask | $34.83 \pm 8.81$ | $5.60 \pm 0.38$ |
| SPAR | $\mathbf{21.76 \pm 1.00}$ | $\mathbf{5.09 \pm 0.10}$ |

**Ablations**. Finally, we conduct ablations to quantify the contribution of each design choice (Table 8). Removing geometric augmentation, the spatial reconstruction objective, or spatial embeddings all leads to comparable performance drops, highlighting their complementary roles. Excluding structural embeddings results in the most severe degradation, underscoring their critical role in modeling node-specific characteristics. We also test alternative masking strategies: Node-Drop Masking (masking entire nodes) reduces performance, while Node-Balanced Masking (ensuring a minimum number of unmasked tokens per node) offers slight gains over random masking. We further vary the mask ratio

(0.85 and 0.5) and observe only minor performance changes relative to the default 0.75, indicating the robustness of the framework.

Table 8: Comparison of the MSE and Distance Error between SPAR and ablations on M3N-VC single-vehicle localization task.

| Method | M3N-VC Single-vehicle Localization | |
| --- | --- | --- |
| | MSE $(m^2)$ ($\downarrow$) | Dist. Err. $(m)$ ($\downarrow$) |
| SPAR | $12.98 \pm 0.11$ | $4.20 \pm 0.07$ |
| w/o Geometric Augmentation in Pretrain | $15.59 \pm 0.56$ | $4.67 \pm 0.04$ |
| w/o Reconstructing Spatial Positions | $14.73 \pm 0.35$ | $4.62 \pm 0.03$ |
| w/o Spatial Positional Embedding | $15.12 \pm 0.58$ | $4.67 \pm 0.07$ |
| w/o Structural Positional Embedding | $22.55 \pm 2.98$ | $5.08 \pm 0.13$ |
| + Node-Drop Masking | $17.71 \pm 3.17$ | $4.82 \pm 0.33$ |
| + Node-Balanced Masking | $12.54 \pm 1.63$ | $4.14 \pm 0.21$ |
| + Mask Ratio 0.85 | $13.89 \pm 2.17$ | $4.35 \pm 0.24$ |
| + Mask Ratio 0.5 | $14.81 \pm 2.57$ | $4.45 \pm 0.31$ |

## B  QUALITATIVE VISUALIZATIONS

To complement the quantitative results, we provide qualitative visualizations that illustrate SPAR's spatial reasoning and representation quality across tasks. These examples highlight its ability to accurately localize targets, distinguish between classes, and learn well-structured embeddings compared to baselines.

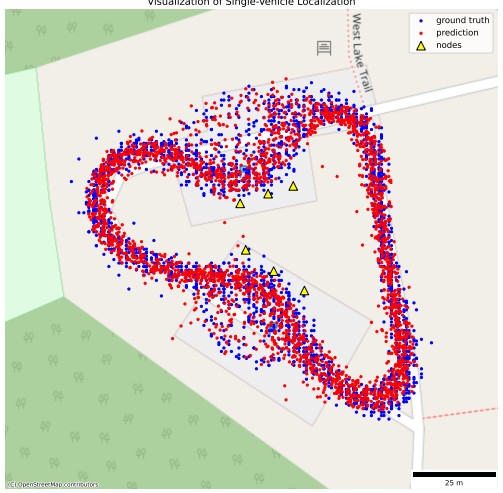
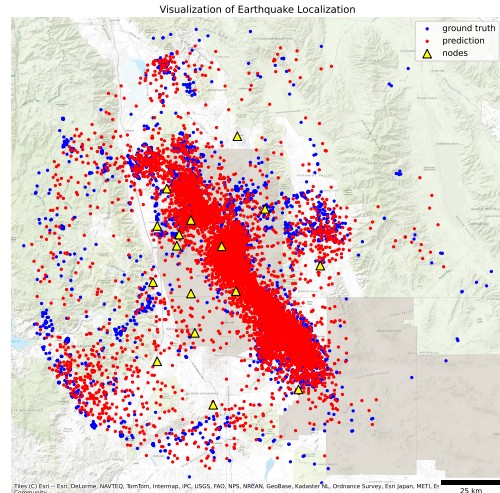

(a) Single-vehicle localization in the M3N-VC dataset, overlaid on an OpenStreetMap basemap (© contributors, ODbL).

(b) Earthquake localization in the Ridgecrest region of California, overlaid on a topographic basemap © Esri, HERE, Garmin, FAO, NOAA, USGS, EPA, NPS.

Figure 3: Visualization of localization results. Blue dots denote ground truth locations (vehicle or earthquake epicenter), red dots are predictions by SPAR, and yellow triangles represent the spatial positions of deployed sensor nodes/stations. SPAR produces predictions that closely align with ground truth locations, demonstrating its robust spatial reasoning capability.

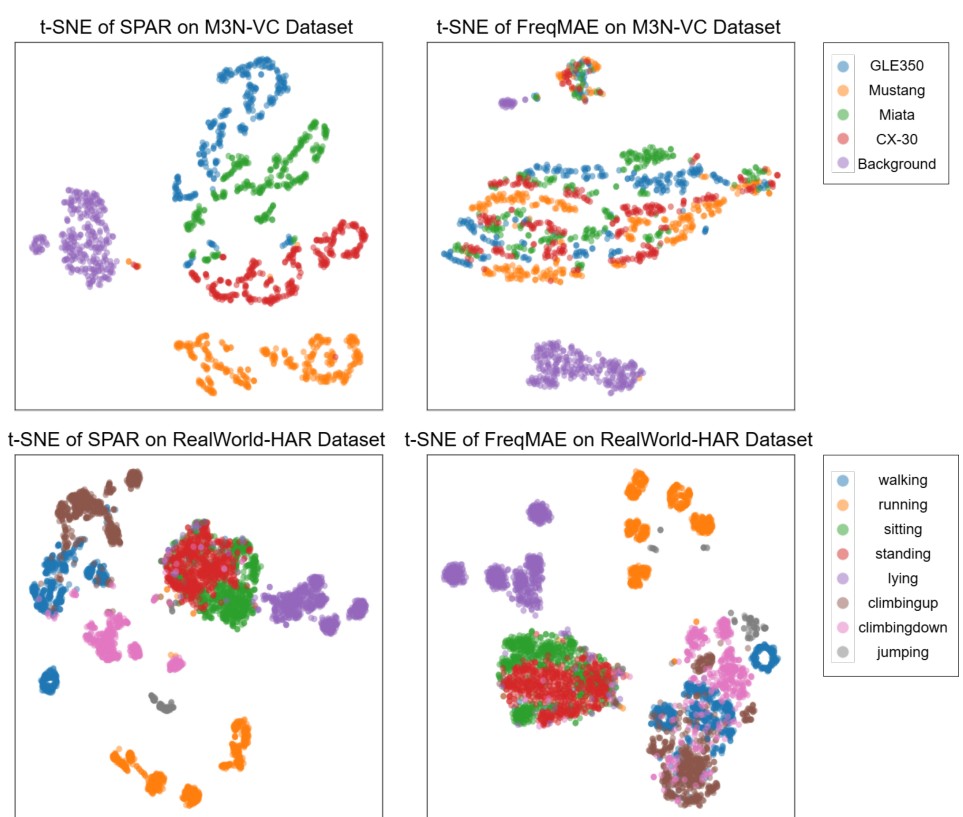

Figure 4: t-SNE plot of SPAR and FreqMAE on the M3N-VC Single-vehicle classification task and on the Realworld-HAR activity recognition task. SPAR produces clearly structured clusters: each vehicle class is distinct and separable from the background, and most activity classes (e.g., Walking, ClimbingUp, ClimbingDown) are well differentiated, with only minor overlap between semantically similar classes like Standing and Sitting. In contrast, FreqMAE yields less structured embeddings, where vehicle classes mix more heavily and activity classes such as Walking, ClimbingUp, and ClimbingDown collapse into broad clusters, indicating weaker fine-grained semantic alignment.

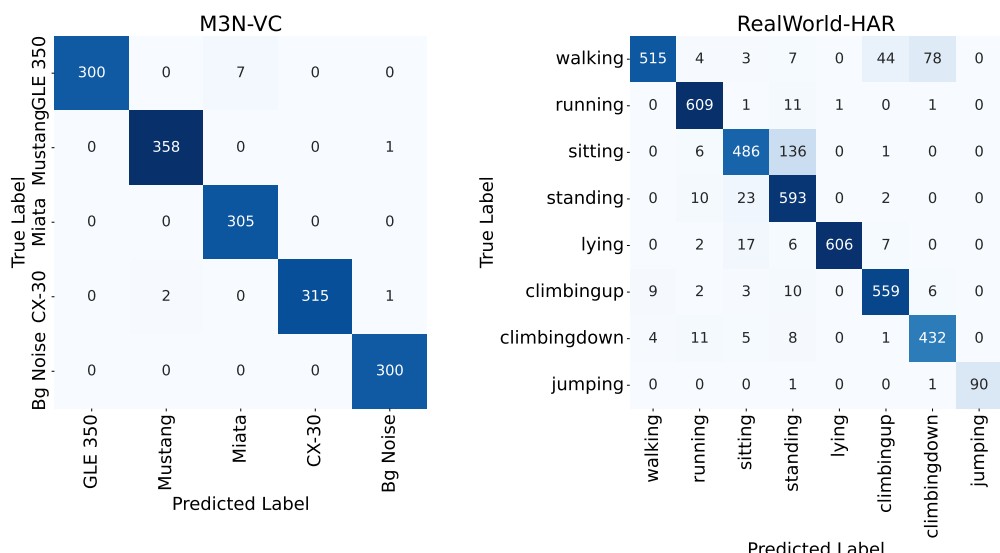

Figure 5: Confusion matrix of SPAR for the M3N-VC single-vehicle classification task (left) and the RealWorld-HAR activity recognition task (right). The classes are mostly separated by SPAR, and the confusion patterns generally align with the conceptual closeness of the classes.

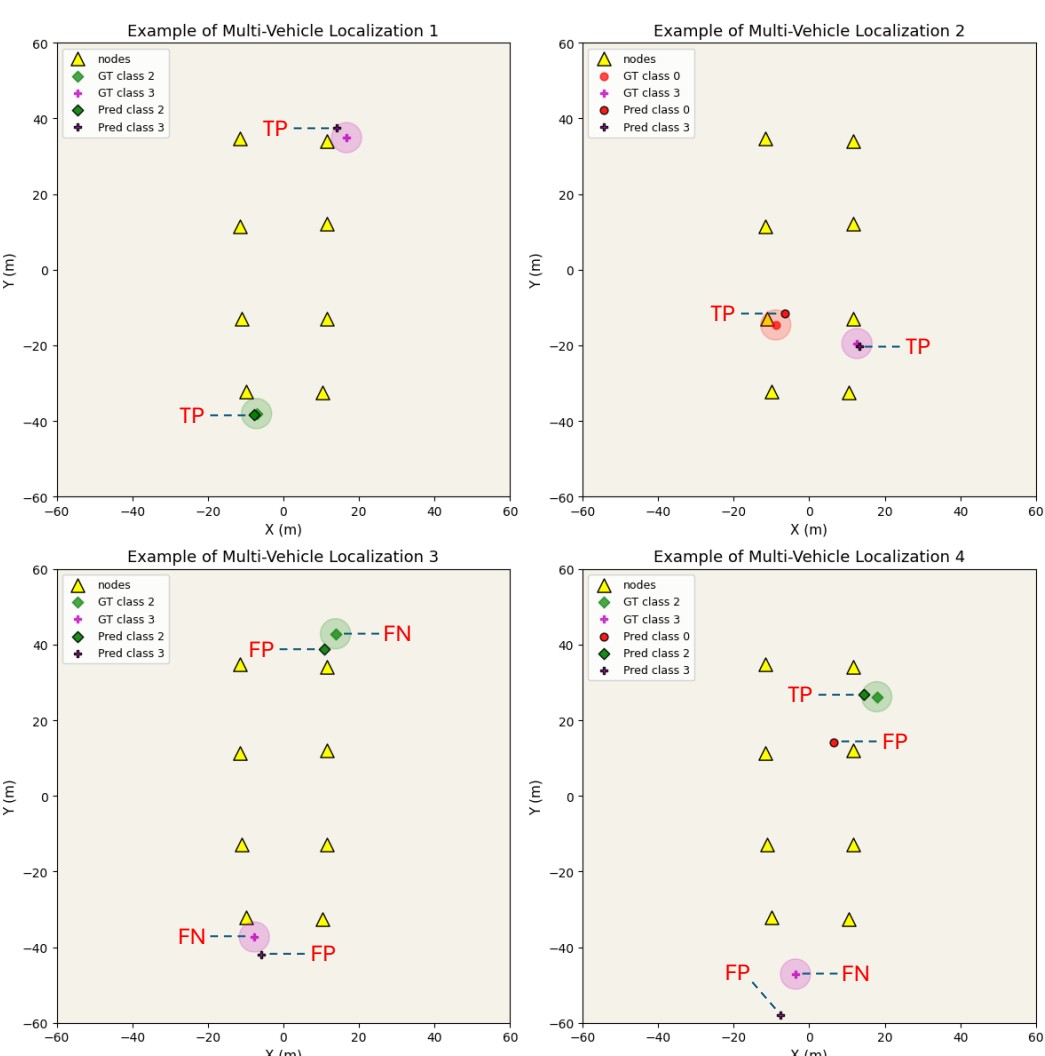

Figure 6: Representative examples from the multi-vehicle localization task. Each subplot displays the ground truth vehicle classes and locations, the predicted classes and locations, and the spatial positions of sensor nodes. A 4-meter radius is drawn around each ground truth vehicle to represent the spatial threshold used for metric mAP@4m during evaluation. Predictions that correctly match both the class and fall within this radius are labeled as true positives (TP). Predictions with incorrect class labels or those that fall outside the threshold are labeled as false positives (FP), while ground truth vehicles with no matching predictions are considered false negatives (FN). The top row shows scenarios where predictions are accurate in both class and location. The bottom row illustrates challenging cases where mismatches in class or location lead to evaluation errors. These illustrations demonstrate SPAR's ability to produce accurate predictions under strict matching criteria.

## C   NOTATION TABLE

For the reader's convenience, we provide a summary of the notations used throughout the paper, along with their corresponding dimensions and definitions, in Table 9.

Table 9: Summary of the notations and their corresponding dimensions and definitions.

| Notation | Dimension(s) | Definition |
|:---:|:---:|:---:|
| $K$ | $\mathbb{N}$ | Number of modalities |
| $n^{(k)}, m^{(k)}, m_{\boldsymbol{M}}^{(k)}$ | $\mathbb{N}$ | Number of nodes, tokens, and masked tokens |
| $d, d_{\boldsymbol{\mathcal{X}}}^{(k)}$ | $\mathbb{N}$ | Model dimension, tokenized signal dimension |
| $d_{\boldsymbol{S}}, d_{\boldsymbol{R}}$ | $\mathbb{N}$ | Spatial and structural position dimensions |
| $L, L'$ | $\mathbb{N}$ | Loss function of SPAR and classical MAE |
| $\boldsymbol{\mathcal{X}}^{(k)}$ | $\mathbb{R}^{n^{(k)} \times m^{(k)} \times d_{\boldsymbol{\mathcal{X}}}^{(k)}}$ | Signals |
| $\widehat{\boldsymbol{\mathcal{X}}}^{(k)}$ | $\mathbb{R}^{n^{(k)} \times m^{(k)} \times d_{\boldsymbol{\mathcal{X}}}^{(k)}}$ | Reconstructed signals |
| $\mathsf{X}^{(k)}$ | $\mathbb{R}^{n^{(k)} \times m^{(k)} \times d_{\boldsymbol{\mathcal{X}}}^{(k)}}$ | Signals random variable |
| $\widetilde{\boldsymbol{\mathcal{X}}}^{(k)}$ | $\mathbb{R}^{n^{(k)} \times m^{(k)} \times d}$ | Signal embeddings |
| $\boldsymbol{S}^{(k)}$ | $\mathbb{R}^{n^{(k)} \times d_{\boldsymbol{S}}}$ | Spatial positions |
| $\boldsymbol{\mathcal{S}}^{(k)}$ | $\mathbb{R}^{n^{(k)} \times m^{(k)} \times d_{\boldsymbol{S}}}$ | Spatial positions (broadcasted) |
| $\widehat{\boldsymbol{\mathcal{S}}}^{(k)}$ | $\mathbb{R}^{n^{(k)} \times m^{(k)} \times d_{\boldsymbol{S}}}$ | Reconstructed spatial positions |
| $\mathsf{S}^{(k)}$ | $\mathbb{R}^{n^{(k)} \times m^{(k)} \times d_{\boldsymbol{S}}}$ | Spatial positions random variable |
| $\widetilde{\boldsymbol{\mathcal{S}}}^{(k)}$ | $\mathbb{R}^{n^{(k)} \times m^{(k)} \times d}$ | Spatial positional embeddings |
| $\boldsymbol{R}^{(k)}$ | $\mathbb{R}^{n^{(k)} \times d_{\boldsymbol{R}}}$ | Structural positions |
| $\boldsymbol{\mathcal{R}}^{(k)}$ | $\mathbb{R}^{n^{(k)} \times m^{(k)} \times d_{\boldsymbol{R}}}$ | Structural positions (broadcasted) |
| $\mathsf{R}^{(k)}$ | $\mathbb{R}^{n^{(k)} \times m^{(k)} \times d_{\boldsymbol{R}}}$ | Structural positions random variable |
| $\widetilde{\boldsymbol{\mathcal{R}}}^{(k)}$ | $\mathbb{R}^{n^{(k)} \times m^{(k)} \times d}$ | Structural positional embeddings |
| $\boldsymbol{M}^{(k)}$ | $\{0,1\}^{n^{(k)} \times m^{(k)}}$ | Mask |
| $\overline{\boldsymbol{M}}^{(k)}$ | $\{0,1\}^{n^{(k)} \times m^{(k)}}$ | Complement mask |
| $\boldsymbol{Z}^{(k)}$ | $\mathbb{R}^{m_{\boldsymbol{M}}^{(k)} \times d}$ | Pre-fusion latent embeddings |
| $\widetilde{\boldsymbol{Z}}^{(k)}$ | $\mathbb{R}^{m_{\boldsymbol{M}}^{(k)} \times d}$ | Post-fusion latent embeddings |
| $\widetilde{\mathsf{Z}}^{(k)}$ | $\mathbb{R}^{m_{\boldsymbol{M}}^{(k)} \times d}$ | Post-fusion latent embeddings random variable |

## D   PROOFS

### D.1   PROOF OF PROPOSITION 3.1

*Proof.* In this proof we use $C$ and $C'$ to denote generic constants independent of model parameters, whose specific values may change from Equation to Equation.

**Classical MAE.** We begin with the case of classical MAE. We assume, following prior works (Li et al., 2022b; Kong & Zhang, 2023), that due to the high dimension of the latent embeddings relative to the original signal, the latent embeddings $\widetilde{\boldsymbol{Z}}^{(k)}$ may contain the full information about the unmasked part of the signals $\mathrm{mask}(\boldsymbol{\mathcal{X}}^{(k)}; \boldsymbol{M}^{(k)})$, which can be reconstructed by the decoder from the latent embeddings with negligible loss. As a result, we consider the reconstruction loss calculated on the

masked signals to be equivalent to the reconstruction loss calculated on the full signals:

$$
\begin{aligned}
L' &= \sum_{k=1}^{K} \|\texttt{mask}(\boldsymbol{\mathcal{X}}^{(k)} - \widehat{\boldsymbol{\mathcal{X}}}^{(k)}; \overline{\boldsymbol{M}}^{(k)})\|_2^2 \\
&= \sum_{k=1}^{K} \|\boldsymbol{\mathcal{X}}^{(k)} - \widehat{\boldsymbol{\mathcal{X}}}^{(k)}\|_2^2 \\
&= \sum_{k=1}^{K} L'^{(k)},
\end{aligned}
\tag{11}
$$

where $L'^{(k)}$ is the reconstruction loss calculated for the $k$-th modality.

Like in the analysis for general regression tasks, the likelihood $P_{\text{dec}}(\mathsf{X}^{(k)}|\widetilde{\mathsf{Z}}^{(k)} = \widetilde{\boldsymbol{Z}}^{(k)})$ implicitly modeled by the decoder is defined as a fully factorized Gaussian distribution with mean $\widehat{\boldsymbol{\mathcal{X}}}^{(k)}$:

$$
P_{\text{dec}}(\mathsf{X}^{(k)}|\widetilde{\mathsf{Z}}^{(k)} = \widetilde{\boldsymbol{Z}}^{(k)}) \stackrel{\text{def}}{=} \mathcal{N}(\widehat{\boldsymbol{\mathcal{X}}}^{(k)}, \frac{1}{\sqrt{2}}\mathrm{I}).
\tag{12}
$$

Then, we can interpret the MSE loss $L'^{(k)}$ as proportional to the negative log-likelihood:

$$
\begin{aligned}
-\log P_{\text{dec}}(\mathsf{X}^{(k)} = \boldsymbol{\mathcal{X}}^{(k)}|\widetilde{\mathsf{Z}}^{(k)} = \widetilde{\boldsymbol{Z}}^{(k)}) &= \|\boldsymbol{\mathcal{X}}^{(k)} - \widehat{\boldsymbol{\mathcal{X}}}^{(k)}\|_2^2 + C' \\
&= L'^{(k)} + C'.
\end{aligned}
\tag{13}
$$

Since the prior probability $P(\mathsf{X}^{(k)} = \boldsymbol{\mathcal{X}}^{(k)})$ is also a constant independent of the model parameters (only determined by the dataset distribution), we can further have

$$
L'^{(k)} + C' = -\log \frac{P_{\text{dec}}(\mathsf{X}^{(k)} = \boldsymbol{\mathcal{X}}^{(k)}|\widetilde{\mathsf{Z}}^{(k)} = \widetilde{\boldsymbol{Z}}^{(k)})}{P(\mathsf{X}^{(k)} = \boldsymbol{\mathcal{X}}^{(k)})}.
\tag{14}
$$

Taking expectation over the data distribution and applying the standard mutual information decomposition, we can have:

$$
\begin{aligned}
\mathbb{E}[L'^{(k)}] + C' &= \mathbb{E}[-\log \frac{P_{\text{dec}}(\mathsf{X}^{(k)}|\widetilde{\mathsf{Z}}^{(k)})}{P(\mathsf{X}^{(k)})}] \\
&= \mathbb{E}[-\log \frac{P(\mathsf{X}^{(k)}|\widetilde{\mathsf{Z}}^{(k)})}{P(\mathsf{X}^{(k)})} + \log \frac{P(\mathsf{X}^{(k)}|\widetilde{\mathsf{Z}}^{(k)})}{P_{\text{dec}}(\mathsf{X}^{(k)}|\widetilde{\mathsf{Z}}^{(k)})}] \\
&= -I(\mathsf{X}^{(k)}; \widetilde{\mathsf{Z}}^{(k)}) + KL(P(\mathsf{X}^{(k)}|\widetilde{\mathsf{Z}}^{(k)})||P_{\text{dec}}(\mathsf{X}^{(k)}|\widetilde{\mathsf{Z}}^{(k)})) \\
&\geq -I(\mathsf{X}^{(k)}; \widetilde{\mathsf{Z}}^{(k)}),
\end{aligned}
\tag{15}
$$

where $P(\mathsf{X}^{(k)}|\widetilde{\mathsf{Z}}^{(k)})$ denotes the non-tractable ground truth conditional distribution determined by the data distribution and the encoders, and $KL(\cdot||\cdot)$ denotes Kullback–Leibler divergence.

Summing over all modalities, we can prove:

$$
-\mathbb{E}[L'] + C' \leq \sum_{k=1}^{K} I(\mathsf{X}^{(k)}; \widetilde{\mathsf{Z}}^{(k)}).
\tag{16}
$$

**SPAR.** For SPAR, the signal decoder takes additional inputs: masked spatial and structural positional embeddings (Equation 3). Let $\mathsf{S}_M^{(k)}$ and $\mathsf{R}_M^{(k)}$ denote the masked spatial and structural positions. Let $L_{\text{sig}}^{(k)}$ denote the signal reconstruction loss for modality $k$. Then, adjusting our reasoning above, we can modify Equation 13 to:

$$
L_{\text{sig}}^{(k)} + C = -\log P_{\text{dec}}(\mathsf{X}^{(k)} = \boldsymbol{\mathcal{X}}^{(k)}|\widetilde{\mathsf{Z}}^{(k)} = \widetilde{\boldsymbol{Z}}^{(k)}, \mathsf{S}_M^{(k)} = \boldsymbol{\mathcal{S}}_M^{(k)}, \mathsf{R}_M^{(k)} = \boldsymbol{\mathcal{R}}_M^{(k)}).
\tag{17}
$$

Since in SPAR, the latent embeddings $\widetilde{\boldsymbol{Z}}^{(k)}$ are calculated not only from unmasked signals, but also from unmasked spatial and structural positional embeddings, we can re-use our assumption above that the latent embeddings $\widetilde{\boldsymbol{Z}}^{(k)}$ retain the full information of them. As a result, we can equivalently condition the likelihood on full spatial and structural positions:

$$
\begin{aligned}
L_{\text{sig}}^{(k)} + C &= -\log P_{\text{dec}}(\mathsf{X}^{(k)} = \boldsymbol{\mathcal{X}}^{(k)}|\widetilde{\mathsf{Z}}^{(k)} = \widetilde{\boldsymbol{Z}}^{(k)}, \mathsf{S}_M^{(k)} = \boldsymbol{\mathcal{S}}_M^{(k)}, \mathsf{R}_M^{(k)} = \boldsymbol{\mathcal{R}}_M^{(k)}) \\
&= -\log P_{\text{dec}}(\mathsf{X}^{(k)} = \boldsymbol{\mathcal{X}}^{(k)}|\widetilde{\mathsf{Z}}^{(k)} = \widetilde{\boldsymbol{Z}}^{(k)}, \mathsf{S}^{(k)} = \boldsymbol{\mathcal{S}}^{(k)}, \mathsf{R}^{(k)} = \boldsymbol{\mathcal{R}}^{(k)}).
\end{aligned}
\tag{18}
$$

As the reasoning above, since the prior probability $P(\mathsf{X}^{(k)} = \boldsymbol{\mathcal{X}}^{(k)}|\mathsf{S}^{(k)} = \boldsymbol{\mathcal{S}}^{(k)}, \mathsf{R}^{(k)} = \boldsymbol{\mathcal{R}}^{(k)})$ is also independent of the model parameters, we can adjust Equation 15 to:

$$
\mathbb{E}[L_{\text{sig}}^{(k)}] + C \geq -I(\mathsf{X}^{(k)}; \widetilde{\mathsf{Z}}^{(k)}|\mathsf{S}^{(k)}, \mathsf{R}^{(k)}).
\tag{19}
$$

Since SPAR treats spatial positions symmetrically to signals. We can apply the same reasoning on signal reconstruction loss to the spatial reconstruction loss $L_{\text{sp}}^{(k)}$, yielding:

$$
\mathbb{E}[L_{\text{sp}}^{(k)}] + C \geq -I(\mathsf{S}^{(k)}; \widetilde{\mathsf{Z}}^{(k)}|\mathsf{X}^{(k)}, \mathsf{R}^{(k)}).
\tag{20}
$$

Summing over all modalities and both reconstruction losses, we can prove:

$$
-\mathbb{E}[L] + C \leq \sum_{k=1}^{K} I(\mathsf{X}^{(k)}; \widetilde{\mathsf{Z}}^{(k)}|\mathsf{S}^{(k)}, \mathsf{R}^{(k)}) + I(\mathsf{S}^{(k)}; \widetilde{\mathsf{Z}}^{(k)}|\mathsf{X}^{(k)}, \mathsf{R}^{(k)}).
\tag{21}
$$

$\square$

### D.2 PROOF OF PROPOSITION 3.2

*Proof.* **Classical MAE.** Kong *et al.* (Kong & Zhang, 2023) provided a rigorous interpretation of classical MAE as a special case of contrastive learning, where the positive pair consists of two complementary masked views of the same input signals. For completeness and clarity, we briefly restate their reasoning here using our notation. For clarity, we focus on a single modality by omitting the superscript $(k)$ and the joint encoder $\mathcal{F}_{\text{joint\_enc}}$; the extension to multiple modalities is straightforward.

Let $\mathcal{F}'_{\text{embed\_enc}}$ denote the composition of the embedding layer and encoder in classical MAE, and let $\mathcal{F}'_{\text{dec}}$ denote the decoder. Then, the reconstruction process can be written as:

$$
\widehat{\boldsymbol{\mathcal{X}}} = \mathcal{F}'_{\text{dec}}(\mathcal{F}'_{\text{embed\_enc}}(\texttt{mask}(\boldsymbol{\mathcal{X}}; \boldsymbol{M}))).
\tag{22}
$$

Accordingly, the reconstruction loss of classical MAE can be rewritten as

$$
\begin{aligned}
L' &= \|\texttt{mask}(\boldsymbol{\mathcal{X}} - \widehat{\boldsymbol{\mathcal{X}}}; \overline{\boldsymbol{M}})\|_2^2 \\
&= \|\texttt{mask}(\boldsymbol{\mathcal{X}}; \overline{\boldsymbol{M}}) - \texttt{mask}(\widehat{\boldsymbol{\mathcal{X}}}; \overline{\boldsymbol{M}})\|_2^2 \\
&= \|\texttt{mask}(\boldsymbol{\mathcal{X}}; \overline{\boldsymbol{M}}) - \texttt{mask}(\mathcal{F}'_{\text{dec}}(\mathcal{F}'_{\text{embed\_enc}}(\texttt{mask}(\boldsymbol{\mathcal{X}}; \boldsymbol{M}))); \overline{\boldsymbol{M}})\|_2^2.
\end{aligned}
\tag{23}
$$

Kong *et al.* (Kong & Zhang, 2023) assumes that due to the high dimension of the latent embeddings relative to the original signals, the latent embeddings produced by the $\mathcal{F}'_{\text{embed\_enc}}$ may approximately preserve all the information of the input. This implies the existence of an alternative decoder $\widetilde{\mathcal{F}}'_{\text{dec}}$ that can satisfy:

$$
\texttt{mask}(\boldsymbol{\mathcal{X}}; \overline{\boldsymbol{M}}) \approx \texttt{mask}(\widetilde{\mathcal{F}}'_{\text{dec}}(\mathcal{F}'_{\text{embed\_enc}}(\texttt{mask}(\boldsymbol{\mathcal{X}}; \overline{\boldsymbol{M}}))); \overline{\boldsymbol{M}}),
\tag{24}
$$

where $\widetilde{\mathcal{F}}'_{\text{dec}}$ can be optimized as:

$$
\begin{aligned}
L_{\widetilde{\mathcal{F}}'_{\text{dec}}} &= \|\texttt{mask}(\boldsymbol{\mathcal{X}}; \overline{\boldsymbol{M}}) - \texttt{mask}(\widetilde{\mathcal{F}}'_{\text{dec}}(\mathcal{F}'_{\text{embed\_enc}}(\texttt{mask}(\boldsymbol{\mathcal{X}}; \overline{\boldsymbol{M}}))); \overline{\boldsymbol{M}})\|_2^2 \\
\widetilde{\mathcal{F}}'_{\text{dec}} &= \underset{\widetilde{\mathcal{F}}'_{\text{dec}}}{\arg\min} \, \mathbb{E}[L_{\widetilde{\mathcal{F}}'_{\text{dec}}}].
\end{aligned}
\tag{25}
$$

Using this approximation, the classical MAE loss can be rewritten as:

$$
\begin{aligned}
L' \approx \|\text{mask}(\widetilde{\mathcal{F}}'_{\text{dec}}(\mathcal{F}'_{\text{embed\_enc}}(\text{mask}(\boldsymbol{\mathcal{X}}; \overline{\boldsymbol{M}}))); \overline{\boldsymbol{M}}) \\
- \text{mask}(\mathcal{F}'_{\text{dec}}(\mathcal{F}'_{\text{embed\_enc}}(\text{mask}(\boldsymbol{\mathcal{X}}; \boldsymbol{M}))); \overline{\boldsymbol{M}})\|_2^2.
\end{aligned}
\tag{26}
$$

To draw a connection to contrastive learning, we define the following similarity measure:

$$
\mathcal{G}'(\boldsymbol{Z}_1, \boldsymbol{Z}_2) \stackrel{\text{def}}{=} \|\text{mask}(\widetilde{\mathcal{F}}'_{\text{dec}}(\boldsymbol{Z}_1); \overline{\boldsymbol{M}}) - \text{mask}(\mathcal{F}'_{\text{dec}}(\boldsymbol{Z}_2); \overline{\boldsymbol{M}})\|_2^2.
\tag{27}
$$

Then the classical MAE loss can be rewritten as:

$$
L' \approx \mathcal{G}'(\mathcal{F}'_{\text{embed\_enc}}(\text{mask}(\boldsymbol{\mathcal{X}}; \overline{\boldsymbol{M}})), \mathcal{F}'_{\text{embed\_enc}}(\text{mask}(\boldsymbol{\mathcal{X}}; \boldsymbol{M}))),
\tag{28}
$$

where $\mathcal{F}'_{\text{embed\_enc}}$ is ensured non-trivial by Equation 25.

This reveals the contrastive learning view of classical MAE: $L'$ encourages the encoder $\mathcal{F}'_{\text{embed\_enc}}$ to produce similar latent representations from two complementary masked views of the same input signals:

$$
\begin{bmatrix} \text{mask}(\boldsymbol{\mathcal{X}}; \boldsymbol{M}), & \text{mask}(\boldsymbol{\mathcal{X}}; \overline{\boldsymbol{M}}) \end{bmatrix},
\tag{29}
$$

which explicitly promotes the learning of occlusion-invariant representations in the signal domain.

**SPAR.** We now turn to SPAR. To unify the components used in encoding, we define an extended encoder $\widetilde{\mathcal{F}}_{\text{enc}}$ that encapsulates the signal, spatial, and structural embeddings, along with the encoder $\mathcal{F}_{\text{embed\_enc}}$ and additional pre-decoder inputs:

$$
\widetilde{\mathcal{F}}_{\text{enc}}(\text{mask}(\boldsymbol{\mathcal{X}}; \boldsymbol{M}), \boldsymbol{\mathcal{S}}, \boldsymbol{\mathcal{R}}) \stackrel{\text{def}}{=} \left( \mathcal{F}_{\text{enc}}(\text{mask}(\widetilde{\boldsymbol{\mathcal{X}}} + \widetilde{\boldsymbol{\mathcal{S}}} + \widetilde{\boldsymbol{\mathcal{R}}}; \boldsymbol{M})), \text{mask}(\widetilde{\boldsymbol{\mathcal{S}}} + \widetilde{\boldsymbol{\mathcal{R}}}; \overline{\boldsymbol{M}}) \right).
\tag{30}
$$

By the same logic as for classical MAE, we can assume the existence of a decoder $\widetilde{\mathcal{F}}_{\text{sig\_dec}}$ that reconstructs $\text{mask}(\boldsymbol{\mathcal{X}}; \overline{\boldsymbol{M}})$ almost losslessly from the output of $\widetilde{\mathcal{F}}_{\text{enc}}$:

$$
\text{mask}(\boldsymbol{\mathcal{X}}; \overline{\boldsymbol{M}}) \approx \text{mask}(\widetilde{\mathcal{F}}_{\text{sig\_dec}}(\widetilde{\mathcal{F}}_{\text{enc}}(\text{mask}(\boldsymbol{\mathcal{X}}; \overline{\boldsymbol{M}}), \boldsymbol{\mathcal{S}}, \boldsymbol{\mathcal{R}})); \overline{\boldsymbol{M}}).
\tag{31}
$$

We now define another similarity measure:

$$
\mathcal{G}_{\text{sig}}(\boldsymbol{Z}_1, \boldsymbol{Z}_2) \stackrel{\text{def}}{=} \|\text{mask}(\widetilde{\mathcal{F}}_{\text{sig\_dec}}(\boldsymbol{Z}_1); \overline{\boldsymbol{M}}) - \text{mask}(\mathcal{F}_{\text{sig\_dec}}(\boldsymbol{Z}_2); \overline{\boldsymbol{M}})\|_2^2.
\tag{32}
$$

Let $L_{\text{sig}}$ denote the signal reconstruction loss in SPAR. Then we have the approximation similar to Equation 28:

$$
L_{\text{sig}} \approx \mathcal{G}_{\text{sig}}(\widetilde{\mathcal{F}}_{\text{enc}}(\text{mask}(\boldsymbol{\mathcal{X}}; \overline{\boldsymbol{M}}), \boldsymbol{\mathcal{S}}, \boldsymbol{\mathcal{R}}), \widetilde{\mathcal{F}}_{\text{enc}}(\text{mask}(\boldsymbol{\mathcal{X}}; \boldsymbol{M}), \boldsymbol{\mathcal{S}}, \boldsymbol{\mathcal{R}})).
\tag{33}
$$

Following the same argument of Kong *et al.* (Kong & Zhang, 2023), this shows that $L_{\text{sig}}$ in SPAR can be viewed as a contrastive loss between two masked views of the signal, enriched with shared spatial and structural context:

$$
\begin{bmatrix} (\text{mask}(\boldsymbol{\mathcal{X}}; \boldsymbol{M}), \boldsymbol{\mathcal{S}}, \boldsymbol{\mathcal{R}}), & (\text{mask}(\boldsymbol{\mathcal{X}}; \overline{\boldsymbol{M}}), \boldsymbol{\mathcal{S}}, \boldsymbol{\mathcal{R}}) \end{bmatrix}.
\tag{34}
$$

Since SPAR treats spatial positions symmetrically with signals—both in embedding and reconstruction—we can apply the same reasoning to the spatial reconstruction loss $L_{\text{sp}}$. This yields another type of contrastive pair:

$$
\begin{bmatrix} (\boldsymbol{\mathcal{X}}, \text{mask}(\boldsymbol{\mathcal{S}}; \boldsymbol{M}), \boldsymbol{\mathcal{R}}), & (\boldsymbol{\mathcal{X}}, \text{mask}(\boldsymbol{\mathcal{S}}; \overline{\boldsymbol{M}}), \boldsymbol{\mathcal{R}}) \end{bmatrix}.
\tag{35}
$$

$\square$

## E  ADDITIONAL EXPERIMENTAL SETUP

### E.1  BASELINE METHODS DESCRIPTIONS

Below, we provide detailed elaborations on the baseline methods introduced in Section 4.

**CMC** (Tian et al., 2020) Learns shared representations by maximizing mutual information between views, enabling view-agnostic and scalable contrastive learning across multiple modalities.

**Cosmo** (Ouyang et al., 2022) Integrates contrastive feature alignment with attention-based selective fusion to effectively capture shared and distinctive patterns from multimodal data under scarce labeling.

**SimCLR** (Chen et al., 2020) Forms discriminative visual embeddings by aligning augmented image pairs through a nonlinear projection and optimizing the NT-Xent contrastive loss.

**AudioMAE** (Huang et al., 2022) Applies masked autoencoding to audio by operating on spectrogram patches, using a Transformer to reconstruct masked regions and capture time-frequency patterns without relying on external modalities.

**CAV-MAE** (Gong et al., 2022) Combines masked autoencoding and contrastive learning in a unified audio-visual framework, using modality-specific encoders and a joint decoder to learn both fused and aligned representations from spectrogram and image patches.

**FOCAL** (Liu et al., 2023) Separates multimodal time-series signals into shared and private latent spaces, enforcing orthogonality and applying contrastive and temporal constraints to capture both modality-consistent and modality-exclusive features.

**FreqMAE** (Kara et al., 2024b) Enhances masked autoencoding for multimodal sensing by incorporating frequency-aware transformers, factorized fusion of shared and private features, and a weighted loss that prioritizes informative frequency bands and high-SNR samples.

**PhyMask** (Kara et al., 2024a) Improves masked autoencoding by adaptively selecting time-frequency patches based on energy and coherence metrics, enabling efficient and informative masking tailored to physical sensing signals.

### E.2  SETTINGS FOR MULTI-MODAL MULTI-NODE VEHICLE CLASSIFICATION DATASET

The Multi-Modality Multi-Node Vehicle Classification Dataset (M3N-VC) (Li et al., 2025) (CC BY 4.0) comprises synchronized audio and vibration recordings of four vehicle types, along with background noise, collected from March 2023 to October 2024. Data were gathered across six distinct real-world scenes, each featuring diverse terrain types (asphalt, dirt, gravel, and concrete) and varying weather conditions (sunny, rainy, and windy).

Each scene is instrumented with a spatially distributed sensor network composed of 6 to 8 nodes. Every node includes a co-located microphone (sampled at 16 kHz) and a geophone (sampled at 200 Hz). Vehicle GPS trajectories were recorded at a rate of 1 Hz. All recordings are segmented into non-overlapping 2-second clips, resulting in a total of 21,694 samples. These clips are transformed into mel-scale spectrograms for model input. GPS coordinates are converted into meter-level spatial positions using the Local Tangent Plane approximation (Agency, 1987).

The dataset follows the official temporal split for training and validation (approximately 3:1). Unless otherwise noted, all models—including ours and the baselines—are pretrained on all six scenes.

We evaluate model performance on three downstream tasks:

**Single-Vehicle Classification.** For this task, we use scene H24 for both fine-tuning and testing. A simple linear classifier is employed as the task head, trained using standard cross-entropy loss.

**Single-Vehicle Localization.** This task also uses scene H24 for fine-tuning and testing. A single transformer layer is used as the task head and optimized with the mean squared error (MSE) loss.

**Multi-Vehicle Joint Classification and Localization.** For multi-vehicle settings, we use scene I22, which contains multiple moving vehicles. A two-layer transformer serves as the task head, trained with a DETR-style loss function (Carion et al., 2020) to handle set-based predictions.

Additionally, we conduct a **fine-tuning on unseen placement** experiment, where models are pre-trained on all scenes except H08 and H24 (which share similar placements). We then finetune and evaluate on scene H24.

### E.3 SETTINGS FOR RIDGECREST SEISMICITY DATASET

This dataset contains seismic waveform recordings from 31,452 earthquake events (M > -0.5) occurring between January 1, 2020, and December 31, 2024, within an 80 km radius of (35.9°, -117.6°) in the Ridgecrest region of California. The data collection and processing procedures largely follow the methodology outlined by Si *et al.* (Si et al., 2024).

We obtained the earthquake event catalog by querying the Southern California Seismic Network (SCSN) (California Institute of Technology (Caltech), 1926) via the Southern California Earthquake Data Center (SCEDC) (Center, 2013) online catalog. The selected events include magnitudes higher than -0.5 and depths larger than than -5 km. For each event, we collected three-component (East, North, Vertical) waveform data from 16 stations in the California Integrated Seismic Network, using two modalities: high-gain broadband seismometers and high-gain accelerometers. All data are sampled at 100 Hz and retrieved in miniSEED format from the SCEDC Open Data repository.

For each event, we extract a 30.72-second window from all channels as model input. During preprocessing, we detrend the waveforms and apply the Short-Time Fourier Transform (STFT) to generate spectrograms. A 2-35-Hz band-pass filter is applied to remove low-frequency noise (e.g., oceanic and atmospheric microseisms) and high-frequency instrumental or environmental noise. We convert spatial positions of each station from GPS signals to kilo-meter-level positions using Local Tangent Plane projection (Agency, 1987).

We split the dataset temporally: events from 2020 and 2021 are used for training, while events from 2022 to 2024 form the validation set. This results in 22,360 events in the training set and 9,092 events in the validation set.

For the downstream task of earthquake localization, we employ a two-layer transformer as the task head, optimized using the mean squared error (MSE) loss.

### E.4 SETTINGS FOR REALWORLD-HAR DATASET

The RealWorld Human Activity Recognition (HAR) dataset (Sztyler & Stuckenschmidt, 2016) comprises multi-modal activity signals collected from 15 participants. The dataset captures eight common activity types: walking, sitting, lying, climbing down, running, standing, climbing up, and jumping.

Sensor data were collected from seven body-mounted nodes, located at the chest, forearm, head, shin, thigh, upper arm, and waist. For our study, we focus on three sensing modalities: acceleration, gyroscope, and magnetic field. Due to substantial data loss in the forearm sensor, we exclude that position and retain six body locations for analysis. As the dataset does not provide explicit spatial coordinates, we manually assign approximate 3D spatial positions to each sensor based on standard anatomical placement on a standing person.

All sensor signals are resampled to 50Hz and segmented into non-overlapping 4-second windows, resulting in a total of 13,351 samples. To evaluate generalization to unseen individuals, we adopt a subject-based split: data from the first 10 participants are used for training, while data from the remaining 5 participants form the validation set.

For the downstream task of human activity recognition, we use a simple linear layer as the task head, trained with standard cross-entropy loss.

## F LLM USAGE STATEMENT

LLMs were used in a supportive role for polishing the writing and providing occasional coding assistance, with all outputs carefully verified by the authors. Technical ideas, experimental designs, and theoretical analyses were developed by the authors. The authors take full responsibility for the final content of this paper.

