# OpenReview forum: "SPAR: Self-supervised Placement-Aware Representation Learning for Distributed Sensing"
_ICLR.cc/2026/Conference — Submitted to ICLR 2026_

### Official Review · Reviewer_mCR7 · 2025-10-31

**Soundness:** 3
**Presentation:** 2
**Contribution:** 3
**Rating:** 6
**Confidence:** 3

**Summary:**

The paper address self-supervised representation learning in distributed sensor networks, where sensors have a particular spatial configuration (e.g. GPS coordinates of sensor locations). Instead of treating the spatial information of each sensor as meta-data, the paper proposes Self-supervised Placement-Aware Representation learning (SPAR), a new method which introduces spatial embeddings into the representations. SPAR achieves this by following a dual Masked Auto-Encoder (MAE) setup: In its encoder, first shared embeddings are computed for both a signal tokens, and for the spatial sensor positions. The embeddings are then summed to single signal+spatial tokens. To learn these embeddings, two separate masked versions of these tokens are created: one where the signal part is masked, and one where the spatial part is masked. The masked tokens are reconstructed to their unmasked equivalents by two separate decoders, again one for the signal part, and one for the spatial part.
Experiments on three benchmarks show that SPAR pre-training improves performance over other pre-training strategies across various distributed sensing tasks.

**Strengths:**

* This idea of using spatial information about the sensor configuration into the embedding space is novel;
* The paper also shows other types of meta-data can be encoded (e.g. via LLM).
* Experiments on three real-world distributed sensing benchmarks show SPAR pre-training significantly outperforms other pre-training strategies.
* The paper does not just present empirical results, but also an information theoretic perspective on how SPAR compares to regular MAE, which gives a better view on SPAR's reconstruction objective.

**Weaknesses:**

## [W1] Methodology
line 319: "SPAR can be interpreted as performing contrastive learning over two types of enriched positive pairs: ... [(mask(X;M),S,R), mask(X;\bar{M}),S,R)] (9)" -> I don't understand why we want contrastive learning between vectors with the same S and R elements. I understand contrastive learning between mask(X;M) and mask(X;\bar{M}), since those are different, but if S and R are per-definition the same for a pair, then why would it help to add these? Same comment goes for Equation (10).

## [W2] Experiments
While the experiments show excellent performance for SPAR, it is not completely clear yet that this is because of just the pre-training with the spatial and structural information. During evaluation, does SPAR also use these additional information sources by embedding and added these to the tokens? And, do the baselines also have access to this same information during evaluation? If yes, then the good performance might be due to the SPAR model using more information during evaluation, rather than having better embeddings per-se. I wonder if other pretrainnig approaches could not reach similar strong results if the spatial and structural embeddings are given to the final model during the fine-tuning stage. In any case, the paper should be clear on what information is accessible at what stages, for each of the methods.

## [W3] Other clarity and structure issues:
* Experiments to various key claims are missing from the main paper; only baseline comparisons on various tasks are included in the paper (though some results can be found in the Appendix). For example, one of the key claims is that SPAR improves generalization, but no generalization experiments are in the paper. Only in the appendix do we find for example generalization across unseen sensor placements (Table 7). Generalization across downstream tasks (line 025) is not shown. Ablation studies of the proposed design choices have unfortunteley  also been delegated to the Appendix, Table 8. The experiments in the main submission should support the main claims (one of the three baseline comparisons could instead move to the Appendix). As it is, the reader is forced to check the Appendix to validate the claims, effectively extending the paper's page limit.
* Contribution (3), validating the proposed method in experiments, isn't really a separate contribution from contribution ①, the method. Experiments are an expected requirement to show that a proposed method is effective, thus part of the first contribution.
* Line 052: "Our design is guided by a core principle: the duality between positions and signals. That is, spatial and structural configurations are not auxiliary metadata to the signals, but stand in an equal and mutually-determining relationship with signals" -> The word "duality" typically means that two concepts (here, "sensor positions" and "sensor signals") originate from the same underlying principle. Clearly, there is a correlation between position and the captured signal, but I'm not sure why this can be called a "duality". I understand SPAR's dual reconstruction objective models it as a duality related to the embedding space, but appears to me more of a modeling choice, rather than a property of reality.


## Minor:
* Line 294, Eq ⑥: please add citation for this equation, or are you claiming this bound analysis on MAE is a novel contribution of this work?
* The signal and spatial embeddings are "summed", I believe, but the phrasing is not so clear. Sometimes it is referred to as "adding" (l204, l233), "combining" (l241), but it wasn't clear if this meant concatenating or summing. Only in Eq(1) did I see it is a literal summation.
* Figures 4 and 5 are referenced out of order.

**Questions:**

* See my question on methodology in Weaknesses: why are positive pairs containing (partially) the same elements (Eq ⑨ and ⑩) useful for contrastive learning?

* See my question on experiments in Weaknesses: can you support that the better performance reported for SPAR is really due to the proposed pre-training strategy, and not because that model has access to more information on the evaluation spatial layout during inference that the other models do not use? For SPAR, do you for evaluation only use the signal embeddings (like the baselines), or also the positional and structural embeddings?

* line 249: "To enable cross-modal interactions, we then apply a joint transformer encoder over the concatenated latent embeddings from all modalities" -> combining all tokens from all sensors together seems like a large computational burden for the transformer architecture. Is this not a computational bottleneck?

* Experiments, line 373 "We pretrain on the full dataset and finetune only the prediction head (a single transformer layer) on scene 'H24,' which contains a single moving vehicle." Details are missing for the evaluation setup: what is the test set if the full dataset has been used for training? How similar or different is it to the training and fine-tuning data? Was the fine-tuning scene H24 also part of the pretraining? Same comment goes for the multi-vehicle setup in line 414.

---

> ### Author Response · Authors · 2025-12-03
>
> **W1**: Why include identical spatial and structural context in contrastive positive pairs?
>
> **A1:** Thank you for this question. A simple toy example helps clarify the role of the shared context: suppose a dataset where each data point is a tuple $(A, B)$, and we define $(A, B = b_1)$ and $(A', B = b_1)$ as positive pairs, while defining $(A, B = b_2)$ and $(A', B = b_2)$ as negative pairs. If we remove $B$, then we only see $(A)$ and $(A')$ and can no longer determine whether the pairs are positive or negative.
>
> In SPAR Equation (9), $(S, R)$ plays the same role as $B$ in this toy example, and $mask(X;M)$, $mask(X;\bar{M})$ correspond to $A$, $A'$ respectively. Equation (10) follows the same logic symmetrically.
>
>
> **W2**: Clarifying whether SPAR’s gain comes from pretraining or extra inputs at inference.
>
> **A2**: Thank you for this important question. To clarify, SPAR uses spatial and structural information at inference time, while baseline methods do not use them. In contrast, baselines use standard numeric positional embeddings (e.g., token indices in transformers), which can also help differentiate the signals from different nodes.
>
> This behavior is intentional and is the core contribution of our work: to explicitly incorporate and maintain the sensor placement information into representation learning for distributed sensing. And we demonstrated that modeling placement as part of the representation is essential for achieving strong performance in distributed sensing tasks.
>
> **W3**: Clarity and Structure issues.
>
> **A3**: We appreciate the feedback and agree that experiments supporting our central claims should be more visible in the main paper. Due to space constraints, we prioritized covering breadth across multiple real-world tasks in the original submission; however, we agree that emphasizing evidence tied directly to our claims will improve clarity. We will reorganize the experimental section accordingly. We also agree with the reviewer that, in the revision, we will reorganize the contribution so that experiments are clearly presented as validation of the method.
>
> For the use of the term “duality”, we agree with the reviewer that we do not intend to claim a fundamental physical equivalence, but rather use this term to describe a modeling principle: positions and signals are mutually informative and jointly define observations in distributed sensing, so that neither should be treated as auxiliary metadata in the design. We will clarify this more explicitly in the paper.
>
>
> **W4**: Minor.
>
> **A4**: Thank you for pointing out these issues.
>
> (1) The bound in Eq. (6) is derived in this work, but we do not present it as a technical novelty, because it follows a standard information-theoretic analysis pattern.
>
> (2) We confirm that signal, spatial, and structural embeddings are summed element-wise, and we will revise wording throughout the paper to reduce ambiguity.
>
> (3) We will correct the figure numbering and references to ensure they appear in the proper order in the revised manuscript.
>
>
> **W5**: Concern about the computational cost of the joint cross-modal transformer.
>
> **A5**: Thank you for raising this point. In practice, the joint transformer is not a major computational bottleneck because the number of sensing nodes and modalities in typical distributed sensing systems (including our datasets) is modest. Therefore, the cross-modal encoder would not incur significantly more computational cost than single-modal encoders.
>
> **W6**: Clarification needed on pretraining, fine-tuning, and test data splits.
>
> **A6**: Thank you for requesting this clarification. In both the single- and multi-vehicle tasks, we first split the data in each scene into a training and a test set. Then we pretrain on data from the training split from all scenes (including H24 and I22). For downstream evaluation, we then fine-tune the task head using the training split of the target scene only, and report results on that scene’s test split. Thus, no test data is used during pretraining or fine-tuning. We will clarify this more explicitly in the revised paper.

---

### Official Review · Reviewer_2RvZ · 2025-11-01

**Soundness:** 3
**Presentation:** 2
**Contribution:** 3
**Rating:** 6
**Confidence:** 2

**Summary:**

This paper introduces SPAR, self-supervised placement-aware representation learning, a pretraining approach for aggregating spatial information and sensor properties into distributed sensing. Following a masked autoencoder (MAE) framework, the signal, spatial information, and sensor properties are combined and given to an encoding module to learn the latent embeddings that optimize the reconstruction objectives. This approach encourages the model to retain signal, position information, and properties in a self-supervised manner. The experimental evaluation of SPAR is conducted with three multi-modal datasets including M3N-VC dataset, Ridgecrest Seismicity dataset, and RealWorld-HAR and SPAR consistently outperforms several state-of-the-art methods.

**Strengths:**

The paper extends the MAE framework by including spatial positional embeddings and structural positional embeddings from the sensor positions and characteristics. Compared to most of the existing approaches, SPAR improves its performance by considering spatial and structural information in the pretraining phase. The approach seems novel, and the results look promising.

**Weaknesses:**

Given that the baseline methods don’t take spatial information into account, the comparisons seem unfair. In the comparisons, the model size/complexity should also be included because a larger model usually outperforms a smaller model, and it is hard to know whether the improvements come from the inclusion of the spatial information. An ablation study will be useful and improve the clarity of the paper. In Table 8, when dropping certain components, how do they impact the model complexity?

**Questions:**

1.	What is the complexity of SPAR? How does it compare to the baselines?
2.	Is SPAR robust to sensor position errors?

---

> ### Author Response · Authors · 2025-12-03
>
> **W1**: Question on computational complexity and efficiency of SPAR compared to baselines.
>
> **A1:** SPAR has essentially the **same asymptotic complexity as all Transformer-based baselines** used in our comparison. The dominant computational cost comes from self-attention in the encoders, which scales as: $\mathcal{O}(N^2)$, where $N$ is the number of tokens.
>
> Because SPAR uses the **same backbone encoder, tokenization strategy, and model width/depth** as the baselines, the encoder cost is identical. The additional dual decoders introduce only marginal overhead because the decoders are lightweight compared to the encoders. Besides, the decoders are used **only during pretraining**, and removed during fine-tuning and inference.
>
>
> **W2**: Question about the robustness of SPAR to inaccurate or noisy sensor positions.
>
> **A2**: Yes, SPAR is explicitly designed to be robust to sensor position errors. As described in our occlusion-invariant representation analysis, the pretraining objective encourages the model to learn representations that remain stable when sensor placement information is partially missing. Empirically, we validate this robustness through experiments on unseen sensor placements (Table 7) and under lossy communication settings (Table 6). These results show that SPAR degrades gracefully when placement information is noisy or incomplete, while consistently maintaining a clear advantage over baseline methods. Together, these findings demonstrate that SPAR is tolerant to realistic positioning noise and does not rely on precise sensor calibration to remain effective in real-world deployment scenarios.

---

### Official Review · Reviewer_oFoY · 2025-11-01

**Soundness:** 2
**Presentation:** 2
**Contribution:** 2
**Rating:** 4
**Confidence:** 3

**Summary:**

This paper focuses on distributed sensing, where multiple spatially distributed sensors jointly observe an environment, a setting common to diverse applications such as vehicle monitoring, human activity recognition, and earthquake localization. The authors propose SPAR, a self-supervised pretraining framework that explicitly incorporates sensor placement into representation learning. Unlike prior methods that treat placement as auxiliary information, SPAR models both spatial location and structural role through positional embeddings, and employs dual reconstruction objectives to recover both sensor measurements and positions. The paper also provides theoretical justification for this design. Experimental evaluations on three multimodal datasets demonstrate that SPAR consistently improves downstream task performance compared to various baseline pretraining approaches.

**Strengths:**

- This paper addresses an interesting but underexplored task in representation learning for distributed sensing systems.

- The explicit incorporation of both spatial and structural sensor information through positional embeddings and dual reconstruction objectives is well motivated and reasonable.

- The experiments demonstrate consistent and significant improvements over diverse baselines across three real-world multimodal datasets.

**Weaknesses:**

- I find the theoretical analysis section somewhat forced. It largely reiterates the problem setup in mathematical form, but does not clearly convey new conclusions or insights. The statement “This encourages the embeddings to be context-aware and jointly informative of both signals and spatial layout, while avoiding memorizing redundant information” is vague and appears to restate an intuitive design goal rather than derive a meaningful theoretical result. Overall, it is not clear how the theoretical analysis directly supports or explains the empirical findings.

- If I understand correctly, the authors pretrain all baseline methods using the same dataset. While this provides a controlled comparison, it may inadvertently undermine some inherent advantages of the baselines, particularly those that do not require spatial location information and are designed to scale to much larger, unlabeled datasets. In practical scenarios, such methods could leverage significantly more data during pretraining and therefore may achieve stronger performance than reported here.
I agree that using the same data and number of epochs ensures fairness in the experimental setup. However, the results may not fully reflect real-world usage, where certain baselines are intended to benefit from abundant and diverse training data. As a consequence, restricting them to the limited dataset available in this work could disproportionately impair their performance. It would be helpful if the paper discussed this limitation or evaluated baselines under a more realistic large-scale pretraining setting.

- The experiments on unseen sensor placements are particularly interesting and reflect a practical deployment scenario, where sensor layouts may change without retraining from scratch. It would be valuable for the authors to provide more detailed studies and analysis on this aspect, as it highlights one of the key advantages of the proposed approach.

- It is unclear what specific insights or conclusions should be drawn from the qualitative visualizations in Figure 3. Additional explanation or interpretation would help clarify their purpose and relevance.

Minor things:
- Line 348: The phrase “on three multiple multi-modal, multi-node distributed sensing datasets” is redundant. The word “multiple” can be removed.

**Questions:**

Here’s a polished version of your questions with clearer wording and tone:

- What are the concrete conclusions of your theoretical analysis?
Please summarize the key takeaways and how they inform the design choices or predicted behaviors of SPAR.

- Can you provide a more practical comparison with baselines?
For methods that do not require spatial locations and can scale to larger unlabeled corpora, include results that reflect this advantage (e.g., larger pretraining sets or different data scales).

- Please expand the studies on unseen sensor placements.
Add deeper analysis (e.g., ablations, varying degrees of placement shift, robustness curves) to demonstrate performance without retraining when sensor layouts change.

- Clarify the qualitative results.
Explain what insights should be drawn from the visualizations (e.g., Figure 3), how they support the claims, and what specific phenomena are being illustrated.

---

> ### Author Response · Authors · 2025-12-03
>
> **W1**: The concrete conclusions, takeaways, and insights of the theoretical analysis. And how they inform design choices and explain empirical results.
>
> **A1**: Thank you for this constructive feedback.
>
> For the **information-theoretic analysis (Prop. 1)**, our goal is to formalize two key takeaways.
>
> (1) Reconstructing spatial positions enforces that spatial information must be contained in the learned representations.
>
> (2) More importantly, the representation only retains the extra information in spatial positions that is not explained by structural positions, thereby reducing redundancy.
> These takeaways directly explain why the learned representations can be exploited more effectively in downstream applications, especially those related to spatial reasoning.
>
> For the **occlusion-invariance analysis (Prop. 2)**, our goal is to show that SPAR enforces invariance not only to signal occlusions (as in classical MAE), but also to spatial occlusions. This additional notion of invariance makes the learned representation robust to noisy or unseen placements, improving the robustness of the learned representation.
>
> We will revise this part to (1) remove vague phrasing, (2) explicitly state these takeaways.
>
> **W2**: Concern about the fairness of baseline comparison under limited pretraining data.
>
> **A2**: We thank the reviewer for this important point and agree that some baselines (e.g., AudioMAE) are designed to benefit from large-scale generic pretraining. To address this, we conducted an additional experiment using a publicly available large-scale pretrained AudioMAE model released on (https://huggingface.co/hance-ai/audiomae
> ) and fine-tuned it on our downstream tasks under the same protocol. Its performance is shown in the table below. Despite being pretrained on orders of magnitude more data, AudioMAE significantly underperformed SPAR. This large gap reflects that our acoustic data are highly domain-specific, so **pretraining directly on the in-domain distributed sensing data already provides a strong and more appropriate initialization for baseline methods**. This result also reinforces that large-scale pretraining does not compensate for the lack of placement modeling.
>
> | Method                          | Vehicle Classification Accuracy (%) | Vehicle Classification F1 (%) | Vehicle Localization MSE (m²) | Vehicle Localization Dist. Err. (m) |
> |---------------------------------|-------------------------------------|-------------------------------|-------------------------------|-------------------------------------|
> | AudioMAE (Large-scale Pretrained, Finetune Only) | 83.89                              | 83.86                        | 101.93                        | 9.67                                |
> | **SPAR (Ours)**                 | **99.27 ± 0.07**                    | **99.26 ± 0.07**              | **12.98 ± 0.11**              | **4.20 ± 0.07**                      |
>
> **W3**: Request for deeper evaluation on generalization to unseen sensor placements.
>
> **A3**: We thank the reviewer for recognizing the importance of generalization to unseen sensor layouts. To further strengthen this point, we conducted a new experiment where we **randomly rotated and translated the sensor locations at inference time**, simulating deployment under unexpected layout changes without retraining. The results are summarized below. The dramatic performance gap shows that placement-agnostic methods collapse under layout shifts, whereas SPAR remains stable by explicitly modeling sensor geometry.
>
> Besides, we kindly refer the reviewer to the evaluations on unseen placements and placement ablations (Tables 7 and 8 in the Appendix).
>
> | Method          | Localization MSE (m²) ↓ | Distance Error (m) ↓ |
> | --------------- | ----------------------- | -------------------- |
> | FreqMAE         | 897.15                  | 36.46                |
> | **SPAR (Ours)** | **13.15**               | **4.24**             |
>
>
>
>
> **W4**: Clarifying the qualitative results of Figure 3.
>
> **A4**: Thank you for raising this point. Figure 3 is included to provide qualitative validation of SPAR’s spatial reasoning ability beyond numerical metrics. In both vehicle and earthquake localization, SPAR’s predictions closely align with ground-truth locations across **complex, irregular sensor layouts** and different spatial scales, where targets are **non-trivial and not spatially biased**. These visualizations demonstrate that SPAR learns coherent global geometry rather than exploiting dataset artifacts or regress-to-mean behavior. We will revise the paper to explicitly include these clarifications.

---

### Official Review · Reviewer_snLA · 2025-11-02

**Soundness:** 3
**Presentation:** 2
**Contribution:** 2
**Rating:** 2
**Confidence:** 4

**Summary:**

This paper presents SPAR, a self-supervised framework for placement-aware representation learning in distributed sensing systems. SPAR integrates spatial and structural positional embeddings along with dual reconstruction objectives to jointly model the relationship between sensor positions and observed signals. A variant of the proposed method uses an LLM as a preprocessing step to improve generalizability.

**Strengths:**

+ This work leverages emerging LLM capabilities as a preprocessing step, combined with multi-objective learning, to address the sensor placement problem.
+ The method is evaluated on several datasets and includes comparisons against multiple prior approaches.
+  The paper also attempts to provide theoretical analysis on the performance bound.

**Weaknesses:**

-  The novelty of this work appears limited. The proposed structural position representation seems to be obtained via standard latent embedding computation, and in the SPAR-LLM variant, the LLM is primarily used as a preprocessing step. Additionally, dual or multi-objective reconstruction is a well-established technique in the literature.

- The method uses a neural network to compute an embedding from spatial positions, yet also manually normalizes those spatial inputs. It is unclear why such normalization is necessary given that a neural network can typically learn appropriate embeddings directly from raw inputs.

-  The concept and interpretation of structural position are not clearly defined. Although the paper provides examples such as "the body part a sensor is attached to" or "the orientation of a directional measurement device," it is unclear how the proposed embedding reliably captures this semantic information. The description suggests that structural position is simply a latent feature space derived from input data.

-  Related to the above point, the experimental section does not clearly specify which datasets involve sensors with meaningful structural attributes. The paper does not analyze how structural position affects performance in scenarios where such information should be relevant.

-  Proposition 3.1 is difficult to follow due to the introduction of many new variables. Additionally, Lines 307-309 following the proposition claim that SPAR promotes embeddings that capture signal information beyond what is explained by structural and spatial cues, and vice versa. However, it is unclear how this claim is supported by Proposition 3.1, or which components of the equation correspond to this reasoning. The connection between the theoretical analysis and the method remains weak.

- The process for obtaining baseline results is not clearly described. Were the results taken directly from the original papers, or were the methods reimplemented by the authors? Some baselines (e.g., CMC’20 and FOCAL’23) show extremely poor performance (e.g., 0.16 and 0.08 vs. 41.57 for the proposed method in Table 2). It would be helpful for the authors to explain why these baselines perform so poorly in this evaluation.

-  The LLM component is evaluated only on the HAR dataset and yields minimal improvement, which does not convincingly demonstrate the benefit of incorporating an LLM.

-  The paper does not report runtime or computational complexity of the proposed method (with and without the LLM component).

**Questions:**

Please see the Weaknesses section.

---

> ### Author Response · Authors · 2025-12-03
>
> **W1**: Questioning the novelty.
>
> **A1**: We respectfully emphasize that the novelty of this work should **not be assessed by examining individual components in isolation**, but by how they are unified into a single, principled framework for placement-aware learning. While each element (learnable embeddings, multi-objective objectives, LLM-as-feature-extractor) may resemble ideas from prior work on the surface, SPAR is the first framework to treat sensor placement itself as a first-class representation learning target, in a symmetrical way with the signals.
>
> Besides, as summarized in Section 2, while multi-objective reconstruction exists in prior works, it was primarily applied on discrete, domain-specific positional targets (e.g., patch indices, sentence order). By contrast, our approach encodes and then reconstructs continuous physical positions of sensor nodes.
>
>
>
>
>
> **W2**: Why normalize spatial positions if a neural network can learn embeddings from raw coordinates?
>
> **A2**: Normalization is used for the same reason it is standard practice in virtually all neural networks: **raw magnitudes matter for optimization**. Our spatial coordinates differ by orders of magnitude across scenes (e.g., meters vs. kilometers, parking lots vs. seismic regions), and feeding unnormalized values causes unstable gradients and slows or destabilizes training. While a network can in principle learn scale invariance, in practice this wastes model capacity and hurts convergence.
>
>
> **W3**: Unclear definition and semantics of “structural position”.
>
> **A3**: Structural position is an abstract concept intended to capture all placement-related factors that are not explained by spatial coordinates alone, such as orientation, mounting style, device type, and body location.
>
> Structural positional embeddings are learned as explicit, node-level vectors that are shared across all samples of a given sensor and are intentionally designed to be low-dimensional, which encourages them to act as compact summaries of each node’s persistent characteristics rather than memorizing individual signals. Moreover, they are the only per-node positional embeddings that are learnable in our framework, which means they necessarily absorb the stable, placement-dependent effects that cannot be explained by spatial coordinates and received signals alone. As a result, these embeddings encode meaningful semantic attributes of placement (e.g., body location, orientation, or sensor role), rather than functioning as generic latent parameters or transient features tied to individual samples.
>
> **W4**: Lack of clarity on where structural position is relevant and whether its impact is analyzed.
>
> **A4**: **All datasets** in our experiments involve sensors with meaningful structural attributes: body-mounted IMUs (RealWorld-HAR), heterogeneous seismic stations and sampling conditions (Ridgecrest), and multi-node sensing units with device- and site-specific characteristics (M3N-VC). We further **explicitly quantify the effect of structural position** in our ablation study (Table 8), where removing structural positional embeddings causes the largest performance degradation among all components. This directly demonstrates that structural position is both relevant in all evaluated scenarios and a critical contributor to SPAR’s performance.
>
> **W5**: Confusion about Proposition 3.1 and how it supports the stated interpretation.
>
> **A5:** Proposition 3.1 is built on **conditional mutual information**, which statistically measures how much information one variable contains about another *after removing what is already explained by a third variable*. In our case,  $I(X; Z \mid S, R)$ quantifies how much information the embedding $Z$ retains about the signal $X$ that is **not already explained by spatial and structural placement**, while $I(S; Z \mid X, R)$ quantifies how much information $Z$ retains about spatial layout that is **not already explained by the signal**. This is exactly the formal meaning of “beyond what is explained by.” Classical MAE, on the other hand, only maximizes $I(X; Z)$, which quantifies how much information $Z$ retains about the signal $X$ marginally.
>
> With this analysis, our goal is to formalize two key takeaways:
>
> (1) Reconstructing spatial positions enforces that spatial information must be contained in the learned representations.
>
> (2) More importantly, the representation only retains the extra information in spatial positions that is not explained by structural positions, thereby reducing redundancy.
>
> These takeaways directly explain why the learned representations can be exploited more effectively in downstream applications, especially those related to spatial reasoning.

---

> ### Author Response · Authors · 2025-12-03
>
> **W6**: Clarification on baseline implementation and poor baseline performance.
>
> **A6**: All baselines were re-evaluated by us using **their official codebases** under **identical training protocols, architectures, and data splits**; no results were copied from original papers. Table 2 reflects performance on **a new task setting that has never been evaluated in prior work: multi-vehicle joint classification and localization**. Methods such as CMC and FOCAL were originally developed for representation alignment or separation, not dense spatial reasoning. Their poor performance is therefore expected: **this task fundamentally requires the representation to encode explicit spatial and cross-node interactions**, which placement-agnostic methods have difficulties to learn. In contrast, SPAR is explicitly designed to model spatial layout and node-specific structure, which directly explains the large performance gap.
>
>
> **W7**: Limited gains from the LLM-based variant (SPAR+LLM).
>
> **A7**: We clarify that the LLM is essentially **not a core dependency of SPAR**, but a demonstration of **extensibility**: SPAR is a complete, effective framework without LLM component. SPAR+LLM is included to show that the framework can **naturally incorporate textual sensor metadata when available** and utilize them to yield performance improvements. Importantly,**the LLM is run only once offline** before training to encode metadata; it is not used during pretraining, fine-tuning, or inference, incurring **negligible computational overhead** in practice.
>
>
> **W8**: Runtime and computational complexity analysis.
>
> **A8**: SPAR has essentially the **same asymptotic complexity as all Transformer-based baselines** used in our comparison. The dominant computational cost comes from self-attention in the encoders, which scales as: $\mathcal{O}(N^2)$, where $N$ is the number of tokens.
>
> Because SPAR uses the same **backbone architecture, tokenization, and model size** as the baselines, the encoder cost is identical. The additional dual decoders introduce only marginal overhead since they are lightweight relative to the encoder and are used **only during pretraining**, not in fine-tuning or inference. For SPAR+LLM, the LLM is executed once offline to encode metadata and is not used during training or inference, introducing **negligible computational overhead**.

---

### Official Review · Reviewer_PBrV · 2025-11-04

**Soundness:** 3
**Presentation:** 2
**Contribution:** 3
**Rating:** 4
**Confidence:** 3

**Summary:**

This paper is set in the context of distributed sensing, with a sender and receiver. I should start with the disclaimer that I am not entirely familiar with the application domain. However, I find the ideas put forth illuminating.

The main premise in the paper is that normally, these applications only account for spatial information (say, in the form of positional embeddings of the sensor). But the authors claim that the structural signature also matters. They go on to show this through a proof which boils down to breaking down the problem in terms of $S$ and $R$. In essence, they device a scheme that is aware of both the spatial and structural embeddings boiling down to the maximization of conditional mutual information, with both factors included.

Formally, they setup a dual objective consisting of:
1. Spatial and structural positional embeddings
2. Reconstruct both signal and spatial positions.

Evaluations are convincing and show that this placement aware formulation works for all the cases considered - vehicle classification, earthquake localization and HAR.

**Strengths:**

+ The formulation isolates a presumably new factor hitherto not considered - the 'structural' information.
+ Evaluations are very convincing. Results bear out in all the cases considered.

**Weaknesses:**

- I understand the proofs and the mathematical language. However, I am totally at a loss as to what these 'structural' factors are. Judging from the paper's language (e.g. "they do not fully represent structural placement
conditions, such as the body part a sensor is attached to, or the orientation used for a directional measurement device (e.g., front-facing versus rear-facing camera on an autonomous car") it looks like a graph connecting relational aspects.
- The above point makes the paper somewhat opaque - especially to outsiders such as myself.

**Questions:**

Please explain what is meant by these structural factors in more detail. I think it warrants a bit of a rewrite.

---

> ### Author Response · Authors · 2025-12-03
>
> **W1 & W2**: What “structural positions” represent in practice, and whether they correspond to a graph or relational structure between sensors.
>
> **A1**: Thank you for raising this important clarification question. By structural positions, we do not mean a graph structure or relational edges between sensors. Rather, structural position is an abstract concept intended to capture **all placement-related factors that are not explained by spatial coordinates alone**.
>
> For example, in the RealWorld-HAR dataset, sensors mounted on the head and chest may be spatially close, yet they produce fundamentally different signals due to differences in body dynamics, orientations, mounting configurations, and hardware characteristics. These latent, node-specific effects are precisely what structural positions are designed to capture.
>
> From an implementation perspective, we do **not explicitly define structural positions as physical variables**. Instead, each node is assigned a **learnable structural positional embedding** (a trainable vector) that is optimized jointly with the model during pretraining. This embedding encodes how a given node systematically observes the environment and serves as an additional input to the transformer encoder alongside the signal and spatial embeddings. In this sense, structural positions act as continuous, learned identifiers that model sensor-specific biases and perspectives.
>
> We will revise the paper to explicitly formalize the definition of structural positions and their corresponding embeddings, and we will add a short discussion to clarify their conceptual and technical distinction from graph-based modeling approaches.

---

### Meta-Review · Area_Chair_zQ1u · 2025-12-24

**Summary:**

This paper proposes SPAR, a placement-aware self-supervised learning framework for distributed sensing with spatial/structural embeddings and dual reconstruction of signals and positions. The problem is relevant and results are promising; the rebuttal adds additional evaluations and clarifications. However, the evidence is insufficient for acceptance: SPAR uses placement inputs at inference while baselines do not, and the paper lacks an explicit apples-to-apples control (e.g., baselines given the same placement inputs, or SPAR ablated to remove them) to attribute gains to pretraining. In addition, “structural position” remains weakly grounded and key ablations plus complexity reporting are not adequately presented in the main paper.

**Reviewer Concerns:**

Reviewer PBrV requested clearer structural semantics; Reviewer snLA questioned novelty, theory–method linkage, baseline clarity, and runtime; Reviewer oFoY asked for clearer theoretical takeaways and stronger comparisons; Reviewers 2RvZ and mCR7 emphasized fairness and controlled ablations. The rebuttal improves clarity but does not resolve the core control/fairness issue.

**Reviewer Scores:**

As the discussion did not lead to any official score updates after the rebuttal, the original scores should be treated as unchanged. The paper initially received 6(2RvZ), 6(mCR7), 4(PBrV), 4(oFoY), and 2(snLA). Even acknowledging that some concerns were partially addressed, the overall ratings and remaining core issues are not sufficient to warrant acceptance.

---

### Decision · Program_Chairs · 2026-01-26

Reject